# More Than Just Functional:
# LLM-as-a-Critique for Efficient Code Generation

**Derui Zhu**[1*]    **Dingfan Chen**[2*]    **Jinfu Chen**[3+]
**Jens Grossklags**[1]    **Alexander Pretschner**[1]    **Weiyi Shang**[4]
[1]Technical University of Munich    [2]Max Planck Institute for Intelligent Systems
[3]Wuhan University    [4]University of Waterloo

## Abstract

Large language models (LLMs) have demonstrated remarkable progress in generating functional code, leading to numerous AI-based coding assistant tools. However, their reliance on the perplexity objective during both training and inference primarily emphasizes functionality, often at the expense of efficiency—an essential consideration for real-world coding tasks. Interestingly, we observed that well-trained LLMs inherently possess knowledge about code efficiency, but this potential remains underutilized with standard decoding approaches. To address this, we design strategic prompts to activate the model's embedded efficiency understanding, effectively using LLMs as *efficiency critiques* to guide code generation toward higher efficiency without sacrificing—and sometimes even improving—functionality, all without the need for costly real code execution. Extensive experiments on benchmark datasets (EffiBench, HumanEval+, COFFE, Mercury) across multiple representative code models demonstrate up to a 70.6% reduction in average execution time and a 13.6% decrease in maximum memory usage, highlighting the computational efficiency and practicality of our approach compared to existing alternatives.

## 1 Introduction

Over recent years, large language models (LLMs) have made significant advances in understanding and generating programming code. These models have enabled a wide range of practical applications, including code generation from natural language descriptions, code completion to assist in real-time development, code translation between programming languages, and code analysis and debugging for identifying and resolving issues in software. Numerous models, such as GPT-4 [1], Claude-3 [2], StarCoder [28, 34], CodeLlama [19], DeepSeek-Coder [20], and Opencoder [24], have showcased strong capabilities in code understanding. These models have become integral components of popular integrated development environments (IDEs), significantly enhancing developer productivity by providing intelligent, context-aware code suggestions. Most existing research and development efforts have predominantly focused on ensuring the correctness of the generated code, often emphasizing functional accuracy (ensuring the generated code meets the intended behavior) and syntactical validity (ensuring adherence to language-specific syntax rules) [21, 27, 40, 31, 32, 9].

In contrast, efficiency—a crucial yet underexplored factor—plays a pivotal role in determining the practicality and sustainability of generated code, as inefficiency can lead to significantly higher computational costs, increased latency, elevated energy consumption, and failures to meet the demands of real-world software development. However, standard LLM training and inference processes lack explicit supervision signals for efficiency, often resulting in generated code that,

39th Conference on Neural Information Processing Systems (NeurIPS 2025).

---

[*]    Equal contribution
[+]    Corresponding author

while functionally correct, lags in execution time and memory usage compared to human-written solutions [43, 37, 22]. For instance, recent results from the EffiBench leaderboard [23] reveal that, on average, LLM-generated code requires 2.59–3.44 times the execution time of human-written solutions, with worst-case execution times up to ~68 times longer. These findings underscore the urgent need for approaches that prioritize efficiency alongside functionality in LLM-generated code.

To fill this gap, a natural solution is to explicitly incorporate knowledge of code execution time and memory complexity into LLM generation and training process. Existing methods include leveraging execution overhead profiles for refinement during inference [22, 9, 38] or fine-tuning models with datasets enriched by performance-improving edits from human programmers [44]. While these techniques have shown promise, they come with inherent limitations: generating execution overhead profiles requires running code in controlled environments, which can be resource-intensive, and obtaining sufficient human-curated data for fine-tuning is challenging.

In response, we propose enhancing the default inference process of code-generating LLMs by explicitly incorporating awareness of code efficiency. Specifically, we introduce a strategically prompted secondary LLM as an efficiency critique, which evaluates the static structure of the abstract syntax tree (AST) of generated code snippets and assigns efficiency-based scores to guide the generation towards producing more efficient outcomes. Unlike existing methods, our approach eliminates the need for additional efficiency-annotated datasets, execution environments, or actual code execution. This makes our solution highly practical, lightweight, and easy to implement.

Despite its simplicity, our proposed method demonstrates significant improvements over existing state-of-the-art approaches. Through extensive experiments on standard code generation benchmarks [23, 7], leveraging various representative language models and diverse metrics, we comprehensively evaluate time and memory efficiency from multiple perspectives. For example, our method achieves an average execution speedup of approximately 17× and a ~19% reduction in normalized execution time compared to existing state-of-the-art results [22].

## 2 Related Work

**LLMs for Code Generation & Refactoring.** The increasing availability of open-source code repositories and advances in training techniques for language models have fueled the rapid growth of code-generating LLMs. Progress in this field spans from models designed specifically for coding tasks, such as AlphaCode [29], CodeGen [36], StarCoder [28, 34], CodeT5 [49], Opencoder [24], DeepSeek-Coder [20], and Qwen2.5-Coder [25], to general-purpose foundation models with code understanding capabilities, such as GPT-4 [1] and Claude-3 [2]. These LLMs have supported various code-related applications, including code generation from description, program repair, automated testing, code translation, type inference, and code summarization. Among these tasks, *code generation from natural-language description* has emerged as a key area for evaluating these models. Although LLMs have achieved notable success on benchmarks such as HumanEval [7] and MBPP [3], their efficiency in terms of execution time and memory usage has received comparatively less attention. Recent studies [43, 37, 22] have revealed that LLM-generated code often lags behind human-written solutions in efficiency, underlining the need for further development. To address this, our work presents a practical approach that significantly enhances the efficiency of LLM-generated code, achieving superior performance and broader applicability compared to existing methods. This focus complements *LLM-driven refactoring* systems, which primarily optimize internal code quality (e.g., readability, modularity, and structural maintainability measured by coupling, cohesion, modularity, and cyclomatic complexity [13, 11, 12]) while restricting algorithm or data-structure changes. In contrast, our setting permits functionality-preserving algorithmic changes and optimizes for measured runtime and memory usage.

**Inference-time Scaling.** Inference-time scaling can effectively enhance LLM performance by strategically allocating computational resources during test time [4, 42, 15, 35, 45, 52]. A variety of methods leverage this principle, which can be broadly categorized into structured reasoning, diverse candidate exploration, and iterative refinement approaches. Structured reasoning techniques, such as *chain-of-thought* [50] and *tree-of-thought* [52, 33] prompting, guide models to articulate intermediate steps, enhancing logical reasoning and interpretability. Similarly, *lookahead search* [17, 53] explores future outcomes to guide current decisions, improving planning and anticipatory capabilities. Diverse candidate exploration methods, such as *best-of-N* [10, 48], produce multiple independent outputs

and select the best one, offering a simple and effective strategy for improving performance. *Beam-search* [16] also fits into this category, maintaining a fixed number of promising candidates at each step to balance exploration and efficiency. Iterative refinement focuses on incrementally improving initial outputs, as in *refinement-based approaches* [35, 9], which make localized adjustments to achieve better results, particularly for tasks that require step-by-step corrections. In our work, we focus on diverse candidate exploration for its simplicity, efficiency, and alignment with our objective of maintaining a large exploration space that encompasses diverse potential solutions.

**Code Efficiency & Performance Analysis.** Performance analysis aims to evaluate the efficiency of the code under various conditions to ensure the code meets the specified performance requirements in real-world applications. Traditionally, performance analysis is categorized into static and dynamic approaches. Static performance analysis detects inefficiencies without executing code by analyzing its structure using techniques such as AST analysis to identify performance anti-patterns [8]. Dynamic performance analysis, on the other hand, measures actual runtime behavior by executing the code, using unit tests [41, 6] and profiling techniques [51] to assess actual runtime behavior and memory usage. Our work mainly leverages LLMs to act as a code performance critique, which eliminates the need for actual execution while providing more expressive and insightful analysis than conventional static methods.

# 3 Method

**Notation.** Let $f_\theta$ denote the target code-generating language model parameterized by $\theta$. We consider a standard code-generation task where the goal is to generate a sequence of code tokens $\boldsymbol{y} = (y_1, y_2, ..., y_L)$ given a task description $\boldsymbol{x}$. The model is trained on a large dataset of code and descriptions by maximizing the conditional likelihood: $f_\theta(\boldsymbol{y}|\boldsymbol{x}) = \prod_{l=1}^{L} f_\theta(y_l|\boldsymbol{y}_{<l}, \boldsymbol{x})$, where $f_\theta(y_l|\boldsymbol{y}_{<l}, \boldsymbol{x})$ predicts the token probabilities given the previous ones and the task description. During standard inference, the model iteratively samples new tokens with $\widehat{y}_l \sim f_\theta(y_l|\boldsymbol{y}_{<l}, \boldsymbol{x})$. To enable controlled diversity, the model's output logits are scaled using a temperature parameter $T$, transitioning the standard likelihood $f_\theta(y_l|\boldsymbol{y}_{<l}, \boldsymbol{x})$ to a temperature-scaled probability distribution $f_\theta^T(y_l|\boldsymbol{y}_{<l}, \boldsymbol{x}) = \frac{f_\theta(y_l|\boldsymbol{y}_{<l}, \boldsymbol{x})^{1/T}}{\sum_{y' \in \mathcal{V}} f_\theta(y'|\boldsymbol{y}_{<l}, \boldsymbol{x})^{1/T}}$, where $\mathcal{V}$ is the vocabulary. Starting with the initial token $y_1$, the model feeds each newly sampled token $\widehat{y}_l$ back into itself to generate the subsequent token $\widehat{y}_{l+1}$, continuing this process until a predetermined stopping criterion is met.

## 3.1 Efficiency-Aware Critique for Decoding

While the standard maximum likelihood objective effectively enables models to fit the distribution of real-world program data, allowing LLMs to mimic human-written code given specific descriptions, it does not explicitly address code efficiency. Although training datasets often include abundant unsupervised or weakly paired examples supporting functional correctness and descriptive alignment, explicit supervision for efficiency is rare. This lack of efficiency-focused signals during training results in limited emphasis on computational or algorithmic optimization. As a result, standard code-generating LLMs frequently underperform in producing efficient code.

To address this limitation, we propose incorporating an ***efficiency-guided critique*** into the generation process via a reward function $r : \bigcup_{l=1}^{\infty} \mathcal{V}^l \to \mathbb{R}$. This function evaluates the efficiency of a generated code segment of length $l$ from vocabulary $\mathcal{V}$ and outputs a scalar value reflecting its efficiency. Integrating this reward function allows the model to prioritize not just functional correctness but also computational efficiency. In the following, we define multiple reward functions, each providing a relatively accurate approximation of code segment efficiency, enabling performance evaluation without the need for time- and resource-intensive code execution.

**Static AST Pattern Matching.** We leverage practical performance-improvement insights at the AST level to define our reward function, explicitly identifying 12 common performance-related patterns from existing software engineering literature [5, 18, 26]. These patterns include "nested loops", "redundant function calls (inside loops)", "redundant function calls (memorization)", "inefficient use of data structures", "excessive function calls in loops", "unnecessary recursion", "deeply nested conditional statements", "inefficient string concatenation", "inefficient file/database operations", "large functions", "inefficient loop terminology", and "potential syntax errors" (see Appendix B for

details). To compute the reward for a code segment $\boldsymbol{y}_{\leq l}$, we first parse the segment into an AST and apply pattern matching to detect the presence of these inefficiencies, applying a predefined penalty for each matched pattern. The final score is then normalized within $[0, 1]$ as follows,

$$r_{\text{AST}}(\boldsymbol{y}_{\leq l}) = 1 - \frac{\sum_{p \in \mathcal{P}} \delta_p \cdot \mathbb{1}\Big[p \in \text{AST}(\boldsymbol{y}_{\leq l})\Big]}{\sum_{p \in \mathcal{P}} \delta_p} \tag{1}$$

where $\delta_p$ represents the penalty associated with pattern $p$ and $\mathbb{1}$ is the indicator function that equals 1 if pattern $p$ is present in the AST of the code segment, and 0 otherwise. This penalization mechanism encourages the model to avoid these common performance pitfalls and guides the decoding search toward more efficient code.

**Prompting LLMs as an Efficiency Critique.** While AST pattern matching as a reward effectively improves generated code efficiency compared to the vanilla LLM baseline (see Table 4), it is limited by its inherent reliance on static feature engineering. As a more scalable alternative, we propose using trained LLMs (themselves) as efficiency critics, leveraging their embedded knowledge of code understanding. Although these models may not autonomously generate desired efficient code without supervision, evaluating the efficiency of code segments and providing critique signals during inference is a comparatively simpler task that they can handle effectively. Specifically, we strategically prompt the model to output a scalar value quantifying the efficiency of a given code segment. We ask the critique LLM to assess factors such as *time complexity*, *space complexity*, *runtime performance*, *memory usage efficiency*, *syntax correctness*, and optionally *AST analysis* and *perplexity* (see Appendix A). Formally, the reward is defined as:

$$r_{\text{LLM}}(\boldsymbol{y}_{\leq l}, \boldsymbol{q}) = f_{\theta_{\text{critique}}}(\boldsymbol{y}_{\leq l}, \boldsymbol{q}) \tag{2}$$

where $f_{\theta_{\text{critique}}}$ denotes the critique LLM, which directly outputs the scalar reward $r_{\text{LLM}}$, and $\boldsymbol{q}$ represents the efficiency-focused prompt.

**General Formulation.** Building upon our consideration of various possible reward functions, we define a general formulation that enhances expressiveness through a linear combination of reward signals:

$$r(\boldsymbol{y}_{\leq l}, \boldsymbol{q}) = \alpha \cdot r_{\text{AST}}(\boldsymbol{y}_{\leq l}) + \beta \cdot r_{\text{LLM}}(\boldsymbol{y}_{\leq l}, \boldsymbol{q}) - \gamma \cdot \text{PP}(\boldsymbol{y}_{\leq l}|f_\theta)$$

where $\text{PP}(\boldsymbol{y}_{\leq l}|f_\theta)$ denotes the perplexity (exponentiated average negative log-likelihood) of code sequence $\boldsymbol{y}_{\leq l}$ given the model $f_\theta$, and $\alpha, \beta, \gamma$ are hyperparameters that modulate the contribution of different reward components. Notably, this formulation flexibly integrates diverse evaluative criteria, enabling the incorporation of future insights into code efficiency for further improvement.

### 3.2 Diverse Candidate Exploration

For decoding search, we adopt widely used inference-time scaling methods that prioritize diverse candidate exploration. This aligns with our objective of generating code that is not only probable but also efficient, as discovering the most optimal solution requires navigating a sufficiently large output space that might otherwise be restricted or biased by a standard perplexity-driven objective. Specifically, we perform perplexity-based ancestral sampling at the ***token level*** until encountering a line break symbol (e.g., \n, \r), which marks the end of a code statement. We then evaluate the reward of each segment at the ***statement level*** and apply *beam search* (with beam width >1) and *greedy search* (i.e., beam search with beam width =1) to expand the search space, following common practice [10, 47, 30, 46, 52]. Finally, we repeat the sampling process multiple times to generate a diverse set of candidates and select the final program by choosing the highest-rewarded one from the candidate set $\mathcal{C}$, formally expressed as: $\boldsymbol{y}^* = \arg\max_{\widehat{\boldsymbol{y}}^{(i)} \in \mathcal{C}} r(\widehat{\boldsymbol{y}}^{(i)}, \boldsymbol{q})$.

## 4 Experiments

### 4.1 Setup

**Datasets, Models, and Hardware.** We conduct experiments on four recent standard code benchmark datasets: **EffiBench** [23], a benchmark comprising 1,000 efficiency-critical LeetCode coding problems paired with human-written canonical solutions, filtered to 988 samples with verified correct

test cases; **HumanEval+** [31], an extension of HumanEval [7] with 164 human-written Python programming tasks with expanded test coverage for rigorous functional correctness evaluation; **Mercury** [14], a dataset of 1,889 Python tasks with test case generators and difficulty annotations derived from solution runtimes; and **COFFE** [39], a code generation benchmark with 398 and 358 problems for function-level and file-level code generation, respectively. Our evaluation includes the following recent open-source code-generating large language models (LLMs) with varying sizes and configurations: **OpenCoder** [24] (the *OpenCoder-8b* checkpoint[1]), **DeepSeekCoder** [20] (i.e., the *DeepSeek-6.7b*[2] and *DeepSeekCoder-v2-16b*[3] checkpoints), **StarCoder** [28] (i.e., *StarCoder2-15b* checkpoint[4]), and **Qwen2.5-Coder** [25] (i.e., the *Qwen2.5-Coder-32B-Instruct*[5] checkpoint). To measure performance, we profile execution time and memory usage using Line Profiler[6] and Memory Profiler[7]. The code generation experiments were conducted on a SLURM-managed computing cluster equipped with 16 NVIDIA A100 Tensor Core GPUs (80GB memory each), interconnected via NVLink 3.0 technology. Each compute node featured 512GB memory. For code performance evaluation, all measurements were conducted in isolated environments. Each test was run on a separate virtual machine instance with identical configurations to minimize system-level variability: a dedicated CPU-only node was deployed containing dual Intel Xeon E5-2695 v4 processors (36 threads total @ 2.1GHz base frequency) with 512GB DDR4-2400 memory. All experiments followed deterministic computing practices with fixed random seeds to ensure reproducibility.

**Metrics.** In line with existing benchmarks [23, 31, 22], we evaluate the efficiency and correctness of the generated code using several key metrics. **Execution Time (ET)** measures the total runtime of the generated code in seconds, while **Normalized Execution Time (NET)** represents the execution time of the generated code divided by that of the canonical human-written solution. **Max Memory Usage (MU)** captures the peak memory consumption (in MB) during execution, with **Normalized Max Memory Usage (NMU)** normalizing this value against the reference solution to assess memory efficiency. **Correctness** is evaluated as the proportion of test samples successfully passing all test cases. When computing execution time and memory usage, we exclude generated code that fails to pass all test cases, ensuring that efficiency metrics are not skewed by severely incorrect programs.

**Baselines & Method Implementation.** We compare our approach against the following baselines, which, to the best of our knowledge, represent the current state-of-the-art: the standard "**Perplexity**"-based generation of LLMs with nucleus (top-p) and best-of-n sampling and selection; "**Self-Debug**" [9], a self-refinement approach aimed at improving code generation correctness; "**EffiLearner**" [22] and "**PerfCodeGen**" [38], which both optimize the efficiency of generated code by incorporating runtime feedback during generation. Before evaluating correctness and performance, we apply a post-processing step to extract the generated code from the model's response without modifying its content. For implementing our methods, we use the prompt shown in Appendix A.2 as default input to the critique LLM, and use the prompt in Appendix A.1 for code generation. For the search strategies, we set a default search space of 50 ($n$=50 for best-of-n) and $p$=0.95 for nucleus (top-p) sampling following common practice [36]. See supplementary materials for more details.

## 4.2 Comparison to Baselines

**Quantitative Results.** In Table 1 (and the supplementary materials), we demonstrate that our method consistently improves both efficiency and correctness metrics across different model architectures with varying model sizes, various datasets and all metrics. Compared to default LLM decoding (i.e., "Perplexity"), our approach reduces the average execution time (ET Avg) by around 92.0%-96.6%, while the median execution time (ET Median) improves by at least 28.2% and up to 58.9%. Additionally, our method lowers memory usage (NMU) by 6.9% to 39.9%. Compared to the previous efficiency-optimized methods, i.e., EffiLearner and PerfCodeGen, our approach achieves substantial reductions in resource usage. Specifically, it reduces average execution time (ET Avg) by

---

1    https://huggingface.co/infly/OpenCoder-8B-Instruct
2    https://huggingface.co/deepseek-ai/deepseek-coder-6.7b-instruct
3    https://huggingface.co/deepseek-ai/DeepSeek-Coder-V2-Lite-Instruct
4    https://huggingface.co/bigcode/starcoder2-15b
5    https://huggingface.co/Qwen/Qwen2.5-Coder-32B-Instruct
6    https://github.com/pyutils/line_profiler
7    https://pypi.org/project/memory-profiler/

| Datasets | Models | Methods | ET(Avg)↓ | ET(Median)↓ | NET(Avg)↓ | NET(Median)↓ | NMU↓ | Correctness↑ |
|---|---|---|---|---|---|---|---|---|
| EffiBench | DeepSeek-6.7b | Perplexity (Best-of-n) | 1.65 | 1.42 | 1.14 | 0.97 | 1.42 | 52.25% |
| | | Perplexity (Top-p) | 1.67 | 1.51 | 1.23 | 1.01 | 1.39 | 50.05% |
| | | Self-Debug | 0.74 | 1.11 | 0.89 | 0.72 | 1.07 | 51.17% |
| | | EffiLearner | 0.68 | 1.02 | 0.71 | 0.69 | 0.94 | 12.40% |
| | | PerfCodeGen | 0.65 | 1.07 | 0.89 | 0.74 | 1.14 | 56.63% |
| | | **Ours** | **0.06** | **0.92** | **0.65** | **0.62** | **0.92** | **57.92%** |
| | OpenCoder-8b | Perplexity (Best-of-n) | 0.73 | 1.38 | 0.84 | 0.94 | 1.05 | 52.41% |
| | | Perplexity (Top-p) | 0.72 | 1.35 | 0.77 | 0.87 | 1.24 | 58.73% |
| | | Self-Debug | 0.72 | 1.41 | 0.87 | 0.89 | 1.16 | 59.78% |
| | | EffiLearner | 0.68 | 1.02 | 0.71 | 0.69 | 0.95 | 12.41% |
| | | PerfCodeGen | 0.64 | 0.87 | 0.69 | 0.71 | 1.01 | 62.88% |
| | | **Ours** | **0.04** | **0.86** | **0.62** | **0.59** | **0.89** | **64.17%** |
| | StarCoder2-15b | Perplexity (Best-of-n) | 0.75 | 1.17 | 0.81 | 0.91 | 1.02 | 41.51% |
| | | Perplexity (Top-p) | 0.77 | 1.17 | 0.81 | 0.91 | 0.97 | 38.23% |
| | | Self-Debug | 0.73 | 1.15 | 0.83 | 0.72 | 0.97 | 51.88% |
| | | EffiLearner | 0.67 | 1.04 | 0.71 | 0.68 | 0.95 | 13.29% |
| | | PerfCodeGen | 0.71 | 1.11 | 0.75 | 0.73 | 1.08 | 50.08% |
| | | **Ours** | **0.06** | **0.84** | **0.62** | **0.56** | **0.95** | **53.81%** |
| | DeepseekCoder-v2-16b | Perplexity (Best-of-n) | 1.39 | 1.41 | 0.79 | 0.83 | 1.05 | 45.51% |
| | | Perplexity (Top-p) | 1.37 | 1.42 | 0.81 | 0.81 | 1.13 | 42.22% |
| | | Self-Debug | 0.71 | 0.74 | 0.81 | 0.83 | 1.04 | 52.37% |
| | | EffiLearner | 0.59 | 0.63 | 0.68 | 0.57 | 1.11 | 14.41% |
| | | PerfCodeGen | 0.65 | 0.71 | 0.77 | 0.79 | 0.97 | 53.32% |
| | | **Ours** | **0.05** | **0.58** | **0.63** | **0.59** | **0.88** | **57.68%** |
| | Qwen2.5-Coder-32b | Perplexity (Best-of-n) | 1.45 | 1.35 | 1.11 | 0.96 | 1.48 | 53.12% |
| | | Perplexity (Top-p) | 1.43 | 1.33 | 1.02 | 0.91 | 1.39 | 50.77% |
| | | Self-Debug | 0.77 | 0.88 | 0.91 | 0.82 | 1.12 | 64.65% |
| | | EffiLearner | 0.62 | 0.73 | 0.76 | 0.68 | 1.08 | 13.23% |
| | | PerfCodeGen | 0.66 | 0.75 | 0.85 | 0.79 | 1.01 | 59.84% |
| | | **Ours** | **0.05** | **0.61** | **0.66** | **0.58** | **0.89** | **66.42%** |
| Mercury | DeepSeek-6.7b | Perplexity (Best-of-n) | 3.55 | 3.71 | 2.84 | 2.98 | 1.11 | 26.14% |
| | | Perplexity (Top-p) | 3.48 | 3.68 | 2.72 | 2.83 | 1.14 | 27.42% |
| | | Self-Debug | 3.74 | 3.86 | 2.91 | 3.09 | 1.01 | 30.01% |
| | | EffiLearner | 3.27 | 3.54 | 2.63 | 2.75 | 1.02 | 8.80% |
| | | PerfCodeGen | 3.21 | 3.09 | 2.49 | 2.56 | 0.97 | 32.07% |
| | | **Ours** | **0.12** | **2.61** | **2.06** | **2.14** | **0.91** | **34.23%** |
| | OpenCoder-8b | Perplexity (Best-of-n) | 3.75 | 3.81 | 2.98 | 3.14 | 1.09 | 27.22% |
| | | Perplexity (Top-p) | 3.69 | 3.77 | 2.85 | 2.94 | 1.13 | 28.81% |
| | | Self-Debug | 3.86 | 3.91 | 3.03 | 3.18 | 1.03 | 31.13% |
| | | EffiLearner | 3.45 | 3.69 | 2.79 | 2.83 | 1.03 | 8.79% |
| | | PerfCodeGen | 3.36 | 3.23 | 2.54 | 2.58 | 0.98 | 33.16% |
| | | **Ours** | **0.13** | **2.74** | **2.13** | **2.24** | **0.93** | **34.88%** |
| | StarCoder2-15b | Perplexity (Best-of-n) | 3.02 | 3.25 | 2.34 | 2.52 | 0.94 | 30.15% |
| | | Perplexity (Top-p) | 2.97 | 3.23 | 2.22 | 2.39 | 0.92 | 30.87% |
| | | Self-Debug | 3.13 | 3.32 | 2.35 | 2.58 | 0.90 | 32.17% |
| | | EffiLearner | 2.70 | 3.02 | 2.08 | 2.23 | 0.88 | 9.32% |
| | | PerfCodeGen | 2.63 | 2.70 | 2.01 | 2.08 | 0.84 | 34.62% |
| | | **Ours** | **0.09** | **2.13** | **1.60** | **1.67** | **0.75** | **37.02%** |
| | DeepseekCoder-v2-16b | Perplexity (Best-of-n) | 3.46 | 3.61 | 2.75 | 2.91 | 1.12 | 27.93% |
| | | Perplexity (Top-p) | 3.41 | 3.59 | 2.64 | 2.78 | 1.11 | 28.65% |
| | | Self-Debug | 3.63 | 3.77 | 2.82 | 3.01 | 1.04 | 30.45% |
| | | EffiLearner | 3.17 | 3.46 | 2.55 | 2.71 | 1.03 | 8.21% |
| | | PerfCodeGen | 3.09 | 3.13 | 2.43 | 2.51 | 0.99 | 32.69% |
| | | **Ours** | **0.11** | **2.47** | **1.92** | **1.97** | **0.89** | **35.01%** |
| | Qwen2.5-Coder-32b | Perplexity (Best-of-n) | 3.28 | 3.42 | 2.61 | 2.77 | 1.08 | 35.91% |
| | | Perplexity (Top-p) | 3.33 | 3.45 | 2.53 | 2.67 | 1.09 | 36.88% |
| | | Self-Debug | 3.54 | 3.64 | 2.68 | 2.88 | 1.03 | 39.18% |
| | | EffiLearner | 3.14 | 3.39 | 2.46 | 2.59 | 1.01 | 9.52% |
| | | PerfCodeGen | 3.01 | 2.97 | 2.34 | 2.43 | 0.95 | 31.14% |
| | | **Ours** | **0.10** | **2.52** | **1.97** | **2.02** | **0.89** | **43.77%** |

**Table 1:** Comparisons of the generated code efficiency across different datasets. Methods that explicitly optimize for code efficiency are highlighted with a  shaded background . The best performance across all methods is indicated in **bold**.

up to 94.1%, median execution time (ET Median) by up to 24.3%, and normalized memory usage (NMU) by up to 20.7%. These improvements are consistent across models and datasets, with average reductions over all five models on EffiBench of 92.0 (ET Avg), 13.8% (ET Median), and 9.4% (NMU) when compared to EffiLearner, and 92.2% (ET Avg), 15.3% (ET Median), and 12.9% (NMU) when compared to PerfCodeGen. More importantly, while EffiLearner struggles with correctness (achieving below 15% accuracy on many models), our method improves correctness by up to 13.3%, demonstrating its ability to generate not only more efficient but also functionally correct code. Overall, our approach consistently achieves substantial gains on all benchmark datasets in our experiments, surpassing both naive decoding and prior efficiency-optimized methods, providing a more effective solution for optimizing execution time, memory usage, and correctness simultaneously.

**Computation Overhead of Each Method.**    Computation overhead during inference is a critical factor for the practical deployment of efficiency-optimized methods. As shown in Table 2, our method introduces significantly less overhead than existing inference-time scaling approaches, with

| Models | Perplexity (Best-of-n) | Perplexity (Top-p) | Self-Debug | EffiLearner | PerfCodeGen | Ours |
|---|---|---|---|---|---|---|
| DeepSeek-6.7b | 3.08 | 2.51 | 4088.59 | 3212.49 | 4144.29 | 28.39 |
| OpenCoder-8b | 2.83 | 2.31 | 3887.27 | 3009.78 | 3987.53 | 26.79 |
| StarCoder2-15b | 5.71 | 4.52 | 6910.46 | 5316.53 | 6985.29 | 121.32 |
| DeepseekCoder-v2-16b | 6.34 | 5.02 | 8177.13 | 6242.98 | 8288.58 | 56.78 |
| Qwen2.5-Coder-32b | 11.41 | 9.18 | 21661.09 | 17145.45 | 22180.94 | 101.34 |

**Table 2:** The *median processing time* (in seconds) of each method on the EFFIBENCH dataset.

processing times that are two to three orders of magnitude lower than those of baseline methods such as `Self-Debug`, `EffiLearner`, and `PerfCodeGen`. In contrast to these baselines—which incur significant computational costs due to repeated and extensive execution of generated code during inference, as well as the considerable engineering effort required to maintain real-time execution environments—our method remains lightweight and efficient. For instance, while `EffiLearner` and `PerfCodeGen` require several thousand seconds to refine the generated outputs from large models like StarCoder2-15b and Qwen2.5-Coder-32b, our method completes inference in 121.32s and 101.34s, respectively. Notably, it is expected that standard `Perplexity`-based decoding (the shaded region marked in Table 2) exhibits minimal overhead, as it does not involve any additional evaluation or optimization steps. However, among all efficiency-aware methods, our approach stands out as significantly more practical, delivering strong performance gains without incurring the substantial computational costs typically associated with code execution-based evaluation loops.

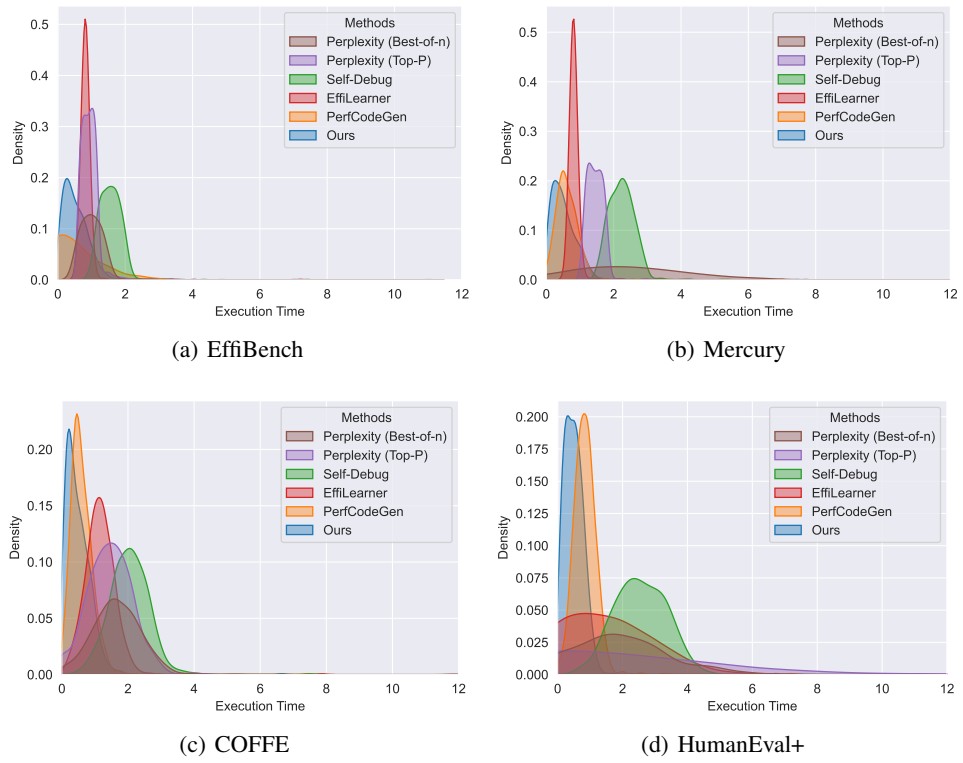

(a) EffiBench      (b) Mercury

(c) COFFE      (d) HumanEval+

**Figure 1:** Execution time (ET) distribution of generated code by OpenCoder-8b across EFFIBENCH, MERCURY, COFFE and HUMANEVAL+.

**Execution Time Distributions of Generated Code.** We further analyze the execution time distributions of the generated code across different methods on the test tasks from various benchmarks. This provides a more comprehensive view of execution efficiency beyond the average and median statistics presented in Table 1. As shown in Figure 1, our method, alongside `EffiLearner` and `PerfCodeGen`, exhibits a substantial reduction in overall execution time compared to the default (`Perplexity`-based) decoding, highlighting the benefits of efficiency-aware approaches. Notably, while refinement-based methods like `Self-Debug` adopt similar post-hoc correction procedures to

EffiLearner and PerfCodeGen, they do not explicitly model efficiency as an optimization objective. As a result, Self-Debug shows little improvement in execution efficiency, underscoring the necessity of explicitly incorporating efficiency considerations into the generation process. Moreover, while default Perplexity-based decoding often leads to low-efficiency code with a heavy-tailed distribution, our efficiency-aware method concentrates the majority of the generated code within shorter execution times, effectively reducing the occurrence of excessively slow outputs. This suggests that the improvements observed in average execution time are not merely superficial artifacts of outliers but reflect a consistent and systematic enhancement in efficiency. Furthermore, our approach demonstrates a more concentrated and effective improvement compared to the EffiLearner and PerfCodeGen baseline, reinforcing its advantages in optimizing execution performance.

```python
def findMedianSortedArrays(self, nums1: List[int], nums2: List[int]) -> float:
    # Ensure nums1 is the smaller array
    if len(nums1) > len(nums2):
        nums1, nums2 = nums2, nums1
    x, y = len(nums1), len(nums2)
    low = 0
    high = x
    while low <= high:
        partitionX = (low + high) // 2
        partitionY = (x + y + 1) // 2 - partitionX
        maxLeftX = float('-inf') if partitionX == 0 else nums1[partitionX - 1]
        minRightX = float('inf') if partitionX == x else nums1[partitionX]
        maxLeftY = float('-inf') if partitionY == 0 else nums2[partitionY - 1]
        minRightY = float('inf') if partitionY == y else nums2[partitionY]
        if maxLeftX <= minRightY and maxLeftY <= minRightX:
            if (x + y) % 2 == 0:
                return (max(maxLeftX, maxLeftY) + min(minRightX, minRightY)) / 2
            else:
                return max(maxLeftX, maxLeftY)
        elif maxLeftX > minRightY:
            high = partitionX - 1
        else:
            low = partitionX + 1
```

$r_{\text{LLM}} = 0.81, r_{\text{AST}} = 1, \text{PP} = 7.23, \text{ET}: 6.45 \times 10^{-4}\text{s}.$

**Figure 2:** OpenCoder generated solution.

```python
def findMedianSortedArrays(self, nums1: List[int], nums2: List[int]) -> float:
    total_length = len(nums1) + len(nums2)
    nums1.append(float('inf'))
    nums2.append(float('inf'))
    (mid1, mid2) = (total_length // 2, (total_length - 1) // 2)
    i = j = 0
    merged = []
    for _ in range(total_length):
        if nums1[i] < nums2[j]:
            merged.append(nums1[i])
            i += 1
        else:
            merged.append(nums2[j])
            j += 1
    return (merged[mid1] + merged[mid2]) / 2
```

$r_{\text{LLM}} = 0.43, r_{\text{AST}} = 1, \text{PP} = 9.27, \text{ET}: 1.52 \times 10^{-3}\text{s}.$

**Figure 3:** Alternative candidate solution 1.

```python
def findMedianSortedArrays(self, nums1, nums2):
    nums = sorted(nums1 + nums2)
    length = len(nums)
    if length % 2 == 0:
        return (nums[length // 2 - 1] + nums[length // 2]) / 2
    else:
        return nums[length // 2]
```

$r_{\text{LLM}} = 0.89, r_{\text{AST}} = 1, \text{PP} = 9.05, \text{ET}: 1.64 \times 10^{-4}\text{s}.$

**Figure 4:** Our generated solution.

```python
def findMedianSortedArrays(self, nums1: List[int], nums2: List[int]) -> float:
    (m, n) = (len(nums1), len(nums2))
    if m > n:
        (nums1, nums2, m, n) = (nums2, nums1, n, m)
    if n == 0:
        raise ValueError
    (i_min, i_max, half_len) = (0, m, (m + n + 1) // 2)
    while i_min <= i_max:
        i = (i_min + i_max) // 2
        j = half_len - i
        if i < m and nums2[j - 1] > nums1[i]:
            i_min = i + 1
        elif i > 0 and nums1[i - 1] > nums2[j]:
            i_max = i - 1
        else:
            if i == 0:
                max_of_left = nums2[j - 1]
            elif j == 0:
                max_of_left = nums1[i - 1]
            else:
                max_of_left = max(nums1[i - 1], nums2[j - 1])
            if (m + n) % 2 == 1:
                return max_of_left
            if i == m:
                min_of_right = nums2[j]
            elif j == n:
                min_of_right = nums1[i]
            else:
                min_of_right = min(nums1[i], nums2[j])
            return (max_of_left + min_of_right) / 2.0
```

$r_{\text{LLM}} = 0.78, r_{\text{AST}} = 1, \text{PP} = 8.57, \text{ET}: 5.97 \times 10^{-4}\text{s}.$

**Figure 5:** Alternative candidate solution 2.

## 4.3 Analysis Studies

**Qualitative Examples.** We show in Figures 2–5 a concrete example of the generated candidate solutions for the "Median of Two Sorted Arrays" problem. Specifically, we illustrate the corresponding reward values and the final execution time alongside each code snippet. As observed, the default OpenCoder decoding prioritizes the solution with the lowest perplexity (PP), which often leads to suboptimal efficiency. In contrast, our critique-based LLM reward ($r_{\text{LLM}}$) demonstrates a strong correlation with actual execution time, providing a more effective signal for efficiency evaluation. While the vanilla AST-based reward ($r_{\text{AST}}$) can be useful, it may not always be informative, as seen in this case where the predefined patterns do not directly apply. To address this, we incorporate AST analysis into the LLM critique prompts, enabling a more adaptive and expressive evaluation that benefits from both structural insights and learned assessments.

**Varying Critique LLM.** We analyze the impact of using different critique LLMs within our framework, while keeping OpenCoder-8b fixed as the target generation model. As shown in Table 3, larger critique LLMs generally yield better overall performance—achieving lower ET, NET, and NMU values, along with the highest correctness score of 67.73%. This result is expected, as larger models typically encode more domain knowledge, allowing them to more effectively evaluate the

| Critique LLMs | ET(Avg) | NET(Avg) | NMU | Correctness |
|---|---|---|---|---|
| DeepSeek-6.7b | 0.06 | 0.62 | 0.95 | 59.87% |
| OpenCoder-8b | 0.04 | 0.59 | 0.89 | 64.17% |
| DeepseekCoder-v2-16b | 0.04 | 0.56 | **0.88** | 66.49% |
| Qwen2.5-Coder-32b | **0.03** | **0.53** | **0.88** | **67.73%** |

**Table 3:** Comparison of different *critique LLMs* of our method with OpenCoder-8b as the *target generation LLM* on EFFIBENCH.

| Prompt Strategies | NET (Median) | NMU | Correctness |
|---|---|---|---|
| Default (Appendix A.2) | **0.73** | **0.91** | **63.89%** |
| w/o AST (Appendix A.3) | 0.75 | 0.94 | 63.03% |
| w/o perplexity (Appendix A.4) | 0.74 | 0.94 | 63.21% |
| w/o AST and perplexity (Appendix A.5) | 0.74 | 0.94 | 62.88% |

**Table 4:** Comparison of varying prompting strategies for critique LLMs on EFFIBENCH.

efficiency of generated code based on a deeper understanding of program structure and execution behavior. Nonetheless, even when smaller or architecturally different models are used as the critique LLM, the overall trend remains consistent: our method continues to significantly outperform all baselines across key metrics. Although absolute performance may decline slightly—particularly in memory-related metrics—when adopting small critique models, the relative gains over default decoding and previous approaches remain substantial. This indicates that while more powerful critique LLMs can offer additional benefits, our method is not strictly dependent on model size or architectural alignment, and generalizes robustly across different critique configurations.

**Varying Prompting Strategies.** We investigate the impact of different prompting strategies for critique LLMs in Table 4. The results indicate only minor variations in performance, with all variants achieving substantial improvements over the baselines. The best-performing strategy is our default prompt (Appendix A.2), which integrates both AST analysis and perplexity, achieving the lowest NET, lowest NMU, and highest correctness at 63.89%. Removing AST analysis (w/o AST) or perplexity (w/o perplexity) results in slightly worse performance, while omitting both (w/o AST and perplexity) leads to the most noticeable decline. These findings validate the reasonableness of our default choice, suggesting that incorporating multiple perspectives—structural insights from AST analysis and likelihood estimation via perplexity—enhances the critique LLM's effectiveness. Furthermore, the relatively stable results across different prompting strategies highlight the inherent expressive power of LLMs, allowing them to adaptively balance different aspects without requiring fragile or intensive hyperparameter tuning, This flexibility enables the critique LLM to emphasize useful patterns while mitigating potential weaknesses in individual components, making our approach more robust across various settings.

**Reward vs. Runtime Alignment.** We analyze the alignment between the LLM-derived reward used at decoding time with real execution efficiency. Concretely, for each program $i$ we form a paired observation $(r_{\text{LLM}}^i, \Delta t^i)$, where $r_{\text{LLM}}^i$ is the scalar reward assigned by the critique LLM to our method's final generated code, and $\Delta t_i = t_{\text{ours}}^i - t_{\text{baseline}}^i$ is the per-sample runtime difference relative to the corresponding reference baseline (i.e., "Perplexity (best-of-n)"). Using differences rather than raw times controls for program-specific difficulty and isolates efficiency changes attributable to our decoding. We then compute the Pearson correlation coefficient across the set of pairs, i.e., $\rho = \text{corr}(\{r_{\text{LLM}}^i\}, \{\Delta t^i\})$, which represents the correlation between the centered, standardized vectors of $\{r_{\text{LLM}}^i\}$ and $\{\Delta t^i\}$, to quantify how closely the reward signal tracks actual performance gains. Here, values closer to $-1$ indicate stronger alignment, since lower $\Delta t_i$ means faster execution. Averaged over samples from multiple datasets and models, we observe strong negative correlations: $-0.78$ on EFFIBENCH, $-0.69$ on MERCURY, and $-0.74$ on COFFE, suggesting that higher LLM rewards are generally associated with greater reductions in execution time.

# 5 Discussions & Limitations

**Inference-time Scaling.** Our approach builds on the premise that trained LLMs already possess substantial knowledge about code efficiency and have been exposed to diverse programming patterns during pre-training. This enables them to understand code efficiency and generate diverse solutions, including highly optimized ones. This aligns with recent findings that inference-time scaling can be more effective than training-time scaling (i.e., fine-tuning), as it fully utilizes the model's pre-trained knowledge without requiring costly retraining.

In the context of our task, pre-training corpora frequently contain recurring efficiency-related patterns and anti-patterns that LLMs can internalize. Moreover, many online sources included in the training

data, such as developer forums and technical blogs, explicitly discuss runtime behavior or algorithmic complexity alongside code examples. Consequently, models may implicitly learn correlations between code structure and efficiency. While final runtime depends partly on hardware, the dominant determinants of efficiency are algorithmic and structural (which can be captured by the program's computational graph), and can be analyzed through static code representations.

Nonetheless, inference-time scaling remains bounded by the model's pre-existing knowledge and may underperform in domains where such information is scarce. In these cases, fine-tuning on curated datasets of highly optimized code could provide a complementary path to adapt the model toward domain-specific efficiency requirements.

**Practicality, Scalability, and Generality.** Existing decoding strategies prioritize perplexity, which reflects code frequency in training data but does not correlate with execution efficiency, often leading to suboptimal results. We address this by introducing an efficiency-aware ranking mechanism, achieving substantial improvements in execution time, memory usage, and correctness, making the generated code more practical for real-world deployment. At the same time, our method improves the practicability of the generation pipeline by avoiding the computational overhead of executing code during inference, a limitation of existing efficiency-aware approaches. (See additional results in Appendix D.2.) While invoking an LLM as a critique introduces some cost, it offers a favorable trade-off by eliminating the need for direct execution and environment setup. Moreover, preliminary experiments suggest potential efficiency gains by using high-temperature perplexity-based token generation for diversity, followed by parallelized best-of-n selection with efficiency-aware rewards, reducing inference overhead to parallel critique LLM calls on a fixed set of sequences while ensuring efficiency-aware optimization and practical scalability.

Generality across programming languages is also an important aspect of practicality. In principle, our method can be extended to other languages, as high-level AST patterns are often similar and the LLM-based reward component is designed to be language-agnostic, given appropriate training. While some engineering effort is required to adapt the AST extraction module for each target language, this does not present a fundamental obstacle.

Finally, building on its generality, our method also demonstrates practical scalability. The overall processing time grows approximately linearly with code length, which allows it to handle reasonably large code snippets without a significant increase in computational cost. In more complex, multi-file coding problems, additional engineering may be needed to enable cross-file coordination and dependency tracking. Nevertheless, the approach remains feasible up to the model's maximum input capacity, indicating that its scalability is constrained primarily by the backbone model's context size rather than by the design of our generation pipeline.

## 6   Conclusion

In summary, our work enhances the practicability of code-generating LLMs by improving both generated code efficiency and the generation pipeline. We introduce an efficiency-aware sampling mechanism that improves execution time, memory usage, and correctness without requiring costly code execution during inference. Additionally, our analysis of critique LLM variations and prompting strategies demonstrates the flexibility and robustness of our approach across different settings. These findings pave the way for more efficient and scalable LLM-based code generation, with potential for further optimizations and broader real-world adoption.

## Acknowledgments

We thank the anonymous reviewers for their valuable comments and constructive feedback, which have significantly improved the quality of this work. This work is partly supported by the National Natural Science Foundation of China (Grant Nos. 62302347 and U2436205).

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

# Appendix

These supplementary materials include the prompt templates (§A), the details of the static AST patterns (§B), the implementation details (§C), and additional results (§D).

## A   Prompt Templates

### A.1   Generation LLM

> You are a software engineer with Python expertise, and your task is to complete the code with the given prefix. Your generated code should be the optimal in time efficiency and memory usage.
>
> During each generation step, you need to rethink step by step whether your generation is optimal in time efficiency. You need to self-evaluate whether the generated code is the optimal in time efficiency, if it is not optimal, you need to reflection and regenerate it.
>
> There are test examples included in the prompt and you need to analyze it. The completed code needs to be included in a code block. $\{$task$\}$

### A.2   Critique LLM (default)

> $\{$code snippet: ``code`` $\}$
>
> Please rate the above code snippet based on the following performance-related criteria:
>
> 1. Time Complexity (Big-O Notation): Assess the time complexity of the code and assign a score reflecting its efficiency in terms of how the time complexity scales with input size. A score closer to 1 indicates highly efficient code (e.g., $O(\log n)$, $O(n)$), while a score closer to 0 indicates inefficient code (e.g., $O(n^2)$, $O(2^n)$).
> 2. Space Complexity (Big-O Notation): Evaluate the space complexity of the code, considering memory usage for variables and data structures. A score closer to 1 represents minimal space usage (e.g., $O(1)$, $O(n)$), while a score closer to 0 reflects high memory consumption (e.g., $O(n)$).
> 3. Running Time Performance: Provide an estimate of the expected running time for typical input sizes (small, medium, large). Assign a score between 0 and 1 based on the speed of execution, with 1 being the fastest and 0 being the slowest.
> 4. Memory Usage Efficiency: Evaluate how effectively the code uses memory resources, including variable allocations and data structures. A score closer to 1 indicates optimal memory usage, while a score closer to 0 indicates inefficiency or excessive memory consumption.
> 5. AST Analysis for Performance: Perform an Abstract Syntax Tree (AST) analysis to assess the efficiency of the code's structure and operations. The analysis should consider factors like loop depth, redundant expressions, and operator usage. Provide a performance score based on how well-optimized the AST is for execution. A score closer to 1 represents an optimized AST structure, while a score closer to 0 indicates a structure with potential inefficiencies.
> 6. If the code contains syntax error, the final score is 0.
> 7. The perplexity of the code snippet is as low as better.
>
> Output: A single numerical value between 0 and 1 that represents the overall performance score based on the above criteria. No additional text or analysis should be provided. Just the final performance score.

## A.3 Critique LLM (without AST)

{code snippet: ''code'' }

Please rate the above code snippet based on the following performance-related criteria:

1. Time Complexity (Big-O Notation): Assess the time complexity of the code and assign a score reflecting its efficiency in terms of how the time complexity scales with input size. A score closer to 1 indicates highly efficient code (e.g., $O(\log n)$, $O(n)$), while a score closer to 0 indicates inefficient code (e.g., $O(n^2)$, $O(2^n)$).
2. Space Complexity (Big-O Notation): Evaluate the space complexity of the code, considering memory usage for variables and data structures. A score closer to 1 represents minimal space usage (e.g., $O(1)$, $O(n)$), while a score closer to 0 reflects high memory consumption (e.g., $O(n)$).
3. Running Time Performance: Provide an estimate of the expected running time for typical input sizes (small, medium, large). Assign a score between 0 and 1 based on the speed of execution, with 1 being the fastest and 0 being the slowest.
4. Memory Usage Efficiency: Evaluate how effectively the code uses memory resources, including variable allocations and data structures. A score closer to 1 indicates optimal memory usage, while a score closer to 0 indicates inefficiency or excessive memory consumption.
5. If the code contains syntax error, the final score is 0.
6. The perplexity of the code snippet is as low as better.

Output: A single numerical value between 0 and 1 that represents the overall performance score based on the above criteria. No additional text or analysis should be provided. Just the final performance score.

## A.4 Critique LLM (without perplexity)

{code snippet: ''code'' }

Please rate the above code snippet based on the following performance-related criteria:

1. Time Complexity (Big-O Notation): Assess the time complexity of the code and assign a score reflecting its efficiency in terms of how the time complexity scales with input size. A score closer to 1 indicates highly efficient code (e.g., $O(\log n)$, $O(n)$), while a score closer to 0 indicates inefficient code (e.g., $O(n^2)$, $O(2^n)$).
2. Space Complexity (Big-O Notation): Evaluate the space complexity of the code, considering memory usage for variables and data structures. A score closer to 1 represents minimal space usage (e.g., $O(1)$, $O(n)$), while a score closer to 0 reflects high memory consumption (e.g., $O(n)$).
3. Running Time Performance: Provide an estimate of the expected running time for typical input sizes (small, medium, large). Assign a score between 0 and 1 based on the speed of execution, with 1 being the fastest and 0 being the slowest.
4. Memory Usage Efficiency: Evaluate how effectively the code uses memory resources, including variable allocations and data structures. A score closer to 1 indicates optimal memory usage, while a score closer to 0 indicates inefficiency or excessive memory consumption.
5. AST Analysis for Performance: Perform an Abstract Syntax Tree (AST) analysis to assess the efficiency of the code's structure and operations. The analysis should consider factors like loop depth, redundant expressions, and operator usage. Provide a performance score based on how well-optimized the AST is for execution. A score closer to 1 represents an optimized AST structure, while a score closer to 0 indicates a structure with potential inefficiencies.
6. If the code contains syntax error, the final score is 0.

Output: A single numerical value between 0 and 1 that represents the overall performance score based on the above criteria. No additional text or analysis should be provided. Just the final performance score.

## A.5 Critique LLM (without AST, without perplexity)

> {code snippet: ``code`` }
>
> Please rate the above code snippet based on the following performance-related criteria:
>
> 1. Time Complexity (Big-O Notation): Assess the time complexity of the code and assign a score reflecting its efficiency in terms of how the time complexity scales with input size. A score closer to 1 indicates highly efficient code (e.g., $O(\log n)$, $O(n)$), while a score closer to 0 indicates inefficient code (e.g., $O(n^2)$, $O(2^n)$).
> 2. Space Complexity (Big-O Notation): Evaluate the space complexity of the code, considering memory usage for variables and data structures. A score closer to 1 represents minimal space usage (e.g., $O(1)$, $O(n)$), while a score closer to 0 reflects high memory consumption (e.g., $O(n)$).
> 3. Running Time Performance: Provide an estimate of the expected running time for typical input sizes (small, medium, large). Assign a score between 0 and 1 based on the speed of execution, with 1 being the fastest and 0 being the slowest.
> 4. Memory Usage Efficiency: Evaluate how effectively the code uses memory resources, including variable allocations and data structures. A score closer to 1 indicates optimal memory usage, while a score closer to 0 indicates inefficiency or excessive memory consumption.
> 5. If the code contains syntax error, the final score is 0.
>
> Output: A single numerical value between 0 and 1 that represents the overall performance score based on the above criteria. No additional text or analysis should be provided. Just the final performance score.

## B Static AST Patterns

1. **Nested Loops**: The program detects nested loops, which are a potential source of inefficiency due to increased time complexity.

```python
# 1. Detect Nested Loops
nested_loops_penalty = 10  # Larger penalty for nested loops
for node in ast.walk(tree):
  if isinstance(node, ast.For) or isinstance(node, ast.While):
    for other_node in ast.walk(tree):
      if isinstance(other_node, (ast.For, ast.While)) and node != other_node:
        if isinstance(node, ast.For) and isinstance(other_node, ast.For):
          score -= nested_loops_penalty
```

2. **Redundant Function Calls (Inside Loops):** Checks for repeated calls to functions like `expensive_function` within loops and suggests moving them outside the loop.

```python
# 2. Detect Redundant Function Calls Inside Loops
redundant_function_calls_penalty = 8  # Moderate penalty for redundant calls
for node in ast.walk(tree):
  if isinstance(node, (ast.For, ast.While)):
    for stmt in node.body:
      if isinstance(stmt, ast.Expr) and isinstance(stmt.value, ast.Call):
        func_name=stmt.value.func.id if isinstance(stmt.value.func,ast.Name) else ""
        if func_name == "expensive_function":
          score -= redundant_function_calls_penalty
```

3. **Redundant Function Calls (Memorization):** Flags redundant calls to functions with identical arguments and suggests memorization.

```python
# 3. Detect Redundant Function Calls (Memoization Opportunities)
redundant_function_calls_memoization_penalty = 5
function_calls = {}
for node in ast.walk(tree):
  if isinstance(node, ast.Expr) and isinstance(node.value, ast.Call):
    func_name = node.value.func.id if isinstance(node.value.func, ast.Name) else ""
    if func_name == "expensive_function":
      if func_name not in function_calls:
        function_calls[func_name] = set()
      args = tuple(ast.dump(arg) for arg in node.value.args)
      if args in function_calls[func_name]:
        score -= redundant_function_calls_memoization_penalty
```

```
        function_calls[func_name].add(args)
```

4. **Inefficient Use of Data Structures:** Identifies inefficient data structures, like using a list for membership testing.

```
# 4. Detect Inefficient Use of Data Structures (list for membership test)
inefficient_data_structure_penalty = 6
for node in ast.walk(tree):
  if isinstance(node, ast.Expr) and isinstance(node.value, ast.Compare):
    if isinstance(node.value.left,ast.Name) and isinstance(node.value.comparators[0],ast.List):
      score -= inefficient_data_structure_penalty
```

5. **Excessive Function Calls in Loops:** Detects function calls in loops that might be expensive and suggests optimization.

```
# 5. Detect Excessive Function Calls in Loops
excessive_function_calls_penalty = 7
for node in ast.walk(tree):
  if isinstance(node, (ast.For, ast.While)):
    for stmt in node.body:
      if isinstance(stmt, ast.Expr) and isinstance(stmt.value, ast.Call):
        if isinstance(stmt.value.func, ast.Name):
          if stmt.value.func.id in ["expensive_function", "some_expensive_function"]:
            score -= excessive_function_calls_penalty
```

6. **Unnecessary Recursion:** Finds unnecessary recursion and suggests a more efficient iterative approach.

```
# 6. Detect Unnecessary Recursion
unnecessary_recursion_penalty = 12
for node in ast.walk(tree):
  if isinstance(node, ast.FunctionDef):
    if any(isinstance(n, ast.Call) and isinstance(n.func, ast.Name) \
      and n.func.id == node.name for n in ast.walk(node)):
      score -= unnecessary_recursion_penalty
```

7. **Deeply Nested Conditional Statements:** Warns when the conditional logic is too deeply nested, which can affect readability and efficiency.

```
# 7. Detect Deeply Nested Conditional Statements
deeply_nested_conditions_penalty = 4
for node in ast.walk(tree):
  if isinstance(node, ast.If):
    depth = 0
    parent = node
    while isinstance(parent, ast.If):
      depth += 1
      parent = parent.parent if hasattr(parent, 'parent') else None
    if depth > 3:
      score -= deeply_nested_conditions_penalty
```

8. **Inefficient String Concatenation:** Detects inefficient string concatenation inside loops and suggests using the `join()` method.

```
# 8. Detect Inefficient String Concatenation
inefficient_string_concatenation_penalty = 6
for node in ast.walk(tree):
  if isinstance(node, ast.For):
    for stmt in node.body:
      if isinstance(stmt, ast.Expr) and isinstance(stmt.value, ast.BinOp) \
        and isinstance(stmt.value.op, ast.Add):
        if isinstance(stmt.value.left,ast.Str) and isinstance(stmt.value.right,ast.Str):
          score -= inefficient_string_concatenation_penalty
```

9. **Inefficient File/Database Operations:** Flags file and database operations inside loops, which could be optimized by batching or caching.

```
# 9. Detect Inefficient File/Database Operations
inefficient_io_operations_penalty = 10
for node in ast.walk(tree):
  if isinstance(node, ast.With):
    for stmt in node.body:
      if isinstance(stmt, ast.Expr) and isinstance(stmt.value, ast.Call):
        func_name = stmt.value.func.id if isinstance(stmt.value.func, ast.Name) else ""
        if func_name in ["open", "execute", "query"]:
          score -= inefficient_io_operations_penalty
```

10. **Large Functions:** Identifies large functions that could benefit from refactoring for clarity and performance.

```python
# 10. Detect Large Functions (Refactoring Opportunity)
large_function_penalty = 8
for node in ast.walk(tree):
  if isinstance(node, ast.FunctionDef):
    function_size = len(node.body)
    if function_size > 20:  # Arbitrary threshold for large functions
      score -= large_function_penalty
```

11. **Inefficient Loop Terminology:** Identifies inefficient loop constructs like `range(len(data))`.

```python
# 11. Detect Inefficient Loop Terminology (e.g., range(len(data)) vs for item in data)
inefficient_loop_terminology_penalty = 6
for node in ast.walk(tree):
  if isinstance(node, ast.For):
    if isinstance(node.iter, ast.Call) and isinstance(node.iter.func, ast.Name) \
      and node.iter.func.id == "range":
      if isinstance(node.iter.args[0], ast.Call) \
        and isinstance(node.iter.args[0].func,ast.Name) and node.iter.args[0].func.id=="len":
        score -= inefficient_loop_terminology_penalty
```

12. **Potential Syntax Errors:** Identifies syntax errors or incomplete code.

```python
try:
  tree = ast.parse(code)
except SyntaxError as e:
  issues.append(f"SyntaxError detected:{e}. Code might be incomplete. Analyzing partial AST.")
  score -= 20  # Penalize incomplete code
```

# C  Implementation Details

We employ the official open-source implementations of `EffiLearner`[8] and `PerfCodeGen`[9] for evaluation. Since no official implementation is available for `Self-Debug`, we re-implemented it based on the descriptions provided in the original paper. For the standard vanilla `Perplexity`-based decoding, we evaluate commonly used sampling strategies, including top-p and best-of-n for a fair and stable comparison. We set a maximum limit of new tokens to 256 to enforce a stopping criterion for token generation. The temperature is set to 1 by default. Furthermore, we use regular expressions to extract the code portion from the model's response. If multiple code implementations are contained, we only extract and test the first one.

Our method is implemented using beam search and best-of-n selection, with a default beam width $b = 1$ and number of trials $n = 50$. Table 5 summarizes the default hyperparameter settings for the different configurations of our method used in the ablation study. The default configuration, corresponding to the main paper results, uses the composite scoring function $\alpha \cdot r_{\text{AST}} + \beta \cdot r_{\text{LLM}} + \gamma \cdot \text{PP}$ with $b = 1$ and $n = 50$.

| | $\alpha$ | $\beta$ | $\gamma$ |
|---|---|---|---|
| $\alpha \cdot r_{\text{AST}} + \beta \cdot \text{PP}$ $(b = 1, n = 50)$ | 1.2 | 0.4 | – |
| $\alpha \cdot r_{\text{LLM}} + \beta \cdot \text{PP}$ $(b = 1, n = 50)$ | 1.0 | 0.4 | – |
| $\alpha \cdot r_{\text{AST}} + \beta \cdot r_{\text{LLM}} + \gamma \cdot \text{PP}$ $(b = 1, n = 50)$ | 1.3 | 1 | 0.4 |
| $\alpha \cdot r_{\text{AST}} + \beta \cdot \text{PP}$ $(b = 50, n = 1)$ | 1.5 | 0.5 | – |
| $\alpha \cdot r_{\text{LLM}} + \beta \cdot \text{PP}$ $(b = 50, n = 1)$ | 0.9 | 0.5 | – |
| $\alpha \cdot r_{\text{AST}} + \beta \cdot \text{LLM}$ $(b = 50, n = 1)$ | 1.2 | 0.8 | – |

**Table 5:** Weight Configuration for Different Reward Functions.

---

[8]  https://github.com/huangd1999/EffiLearner
[9]  https://github.com/SalesforceAIResearch/perfcodege

# D  Additional Results

## D.1  Experiments on HumanEval+ and COFFE dataset

We present the detailed quantitative results in Table 6, comparing different methods on the HU-MANEVAL+ and COFFE datasets. This serves as a supplementary analysis to Table 1 in the main paper.

| Datasets | Models | Methods | ET(Avg)↓ | ET(Median)↓ | NET(Avg)↓ | NET(Median)↓ | NMU↓ | Correctness↑ |
|---|---|---|---|---|---|---|---|---|
| HumanEval+ | DeepSeek-6.7b | Perplexity (Best-of-n) | 1.39 | 1.26 | 0.89 | 0.85 | 1.25 | 49.89% |
| | | Perplexity (Top-p) | 1.44 | 1.26 | 1.06 | 0.86 | 1.15 | 50.12% |
| | | Self-Debug | 0.73 | 1.20 | 0.89 | 0.95 | 1.17 | 54.76% |
| | | EffiLearner | 0.64 | 1.05 | 0.75 | 0.75 | 1.01 | 13.15% |
| | | PerfCodeGen | 0.72 | 0.84 | 0.73 | 0.72 | 0.97 | 55.81% |
| | | **Ours** | **0.05** | **0.82** | **0.63** | **0.62** | **0.91** | **62.20%** |
| | CodeLlama-7b | Perplexity (Best-of-n) | 2.51 | 2.38 | 2.02 | 1.96 | 1.15 | 48.77% |
| | | Perplexity (Top-p) | 2.55 | 2.41 | 2.14 | 1.94 | 1.13 | 47.95% |
| | | Self-Debug | 2.81 | 2.25 | 1.96 | 1.02 | 1.12 | 52.83% |
| | | EffiLearner | 1.70 | 1.10 | 1.81 | 1.80 | 1.06 | 15.94% |
| | | PerfCodeGen | 1.78 | 1.91 | 1.79 | 1.75 | 1.04 | 54.29% |
| | | **Ours** | **1.54** | **1.89** | **1.71** | **1.68** | **0.95** | **60.37%** |
| | OpenCoder-8b | Perplexity (Best-of-n) | 1.33 | 1.21 | 0.83 | 0.81 | 1.18 | 51.72% |
| | | Perplexity (Top-p) | 1.37 | 1.23 | 0.98 | 0.82 | 1.11 | 52.04% |
| | | Self-Debug | 0.69 | 1.12 | 0.85 | 0.91 | 1.09 | 56.94% |
| | | EffiLearner | 0.60 | 0.98 | 0.68 | 0.70 | 0.95 | 11.08% |
| | | PerfCodeGen | 0.68 | 0.81 | 0.70 | 0.68 | 0.94 | 58.02% |
| | | **Ours** | **0.04** | **0.75** | **0.59** | **0.60** | **0.87** | **64.89%** |
| | CodeLlama-13b | Perplexity (Best-of-n) | 1.65 | 1.33 | 0.96 | 0.90 | 1.31 | 48.12% |
| | | Perplexity (Top-p) | 1.50 | 1.34 | 1.10 | 0.91 | 1.20 | 48.39% |
| | | Self-Debug | 0.78 | 1.22 | 0.93 | 0.99 | 1.19 | 53.41% |
| | | EffiLearner | 0.68 | 1.08 | 0.79 | 0.78 | 1.03 | 14.87% |
| | | PerfCodeGen | 0.75 | 0.89 | 0.77 | 0.73 | 1.01 | 54.92% |
| | | **Ours** | **0.17** | **0.53** | **0.44** | **0.51** | **0.93** | **60.08%** |
| | StarCoder2-15b | Perplexity (Best-of-n) | 1.30 | 1.18 | 0.82 | 0.80 | 1.15 | 52.46% |
| | | Perplexity (Top-p) | 1.35 | 1.20 | 0.95 | 0.83 | 1.08 | 52.71% |
| | | Self-Debug | 0.70 | 1.10 | 0.84 | 0.89 | 1.06 | 57.12% |
| | | EffiLearner | 0.61 | 0.96 | 0.66 | 0.69 | 0.93 | 10.58% |
| | | PerfCodeGen | 0.66 | 0.79 | 0.69 | 0.67 | 0.92 | 58.94% |
| | | **Ours** | **0.03** | **0.73** | **0.58** | **0.59** | **0.86** | **65.31%** |
| | DeepseekCoder-v2-16b | Perplexity (Best-of-n) | 1.21 | 1.09 | 0.76 | 0.74 | 1.04 | 54.31% |
| | | Perplexity (Top-p) | 1.26 | 1.11 | 0.87 | 0.78 | 0.99 | 54.89% |
| | | Self-Debug | 0.65 | 1.02 | 0.78 | 0.84 | 1.00 | 59.36% |
| | | EffiLearner | 0.55 | 0.91 | 0.62 | 0.64 | 0.89 | 9.62% |
| | | PerfCodeGen | 0.61 | 0.75 | 0.65 | 0.63 | 0.88 | 61.24% |
| | | **Ours** | **0.02** | **0.67** | **0.51** | **0.56** | **0.82** | **68.47%** |
| | Qwen2.5-Coder-32b | Perplexity (Best-of-n) | 1.10 | 0.98 | 0.68 | 0.66 | 0.96 | 56.78% |
| | | Perplexity (Top-p) | 1.14 | 1.01 | 0.79 | 0.70 | 0.91 | 57.25% |
| | | Self-Debug | 0.59 | 0.96 | 0.71 | 0.79 | 0.94 | 61.58% |
| | | EffiLearner | 0.49 | 0.85 | 0.58 | 0.60 | 0.84 | 8.21% |
| | | PerfCodeGen | 0.56 | 0.71 | 0.60 | 0.58 | 0.83 | 63.45% |
| | | **Ours** | **0.01** | **0.61** | **0.47** | **0.52** | **0.78** | **69.83%** |
| COFFE | DeepSeek-6.7b | Perplexity (Best-of-n) | 2.15 | 2.31 | 1.64 | 1.78 | 1.28 | 40.12% |
| | | Perplexity (Top-p) | 2.08 | 2.22 | 1.55 | 1.63 | 1.24 | 41.75% |
| | | Self-Debug | 2.26 | 2.33 | 1.72 | 1.84 | 1.21 | 43.19% |
| | | EffiLearner | 1.92 | 2.05 | 1.49 | 1.53 | 1.10 | 11.34% |
| | | PerfCodeGen | 1.75 | 1.89 | 1.38 | 1.45 | 1.06 | 44.66% |
| | | **Ours** | **0.18** | **1.69** | **1.21** | **1.30** | **0.91** | **47.90%** |
| | OpenCoder-8b | Perplexity (Best-of-n) | 1.96 | 2.08 | 1.48 | 1.62 | 1.18 | 41.88% |
| | | Perplexity (Top-p) | 1.91 | 2.03 | 1.39 | 1.55 | 1.13 | 43.50% |
| | | Self-Debug | 2.14 | 2.26 | 1.63 | 1.75 | 1.16 | 44.95% |
| | | EffiLearner | 1.83 | 1.96 | 1.35 | 1.40 | 1.08 | 10.97% |
| | | PerfCodeGen | 1.69 | 1.81 | 1.32 | 1.39 | 1.03 | 46.22% |
| | | **Ours** | **0.17** | **1.60** | **1.15** | **1.22** | **0.88** | **48.77%** |
| | StarCoder2-15b | Perplexity (Best-of-n) | 2.01 | 2.18 | 1.51 | 1.67 | 1.22 | 39.21% |
| | | Perplexity (Top-p) | 1.98 | 2.14 | 1.44 | 1.58 | 1.17 | 40.56% |
| | | Self-Debug | 2.20 | 2.30 | 1.69 | 1.80 | 1.19 | 42.37% |
| | | EffiLearner | 1.78 | 1.93 | 1.36 | 1.41 | 1.05 | 9.95% |
| | | PerfCodeGen | 1.66 | 1.77 | 1.29 | 1.34 | 1.00 | 43.66% |
| | | **Ours** | **0.16** | **1.57** | **1.10** | **1.18** | **0.86** | **46.93%** |
| | DeepseekCoder-v2-16b | Perplexity (Best-of-n) | 2.29 | 2.42 | 1.70 | 1.88 | 1.29 | 37.73% |
| | | Perplexity (Top-p) | 2.20 | 2.35 | 1.62 | 1.76 | 1.25 | 39.00% |
| | | Self-Debug | 2.34 | 2.49 | 1.79 | 1.92 | 1.22 | 40.84% |
| | | EffiLearner | 1.94 | 2.09 | 1.45 | 1.52 | 1.12 | 10.11% |
| | | PerfCodeGen | 1.82 | 1.94 | 1.39 | 1.46 | 1.06 | 42.11% |
| | | **Ours** | **0.19** | **1.65** | **1.23** | **1.29** | **0.90** | **45.88%** |
| | Qwen2.5-Coder-32b | Perplexity (Best-of-n) | 2.42 | 2.55 | 1.83 | 1.94 | 1.31 | 36.27% |
| | | Perplexity (Top-p) | 2.37 | 2.49 | 1.76 | 1.86 | 1.27 | 38.03% |
| | | Self-Debug | 2.46 | 2.59 | 1.87 | 1.95 | 1.24 | 39.96% |
| | | EffiLearner | 2.03 | 2.15 | 1.53 | 1.60 | 1.14 | 10.42% |
| | | PerfCodeGen | 1.91 | 2.01 | 1.43 | 1.50 | 1.08 | 41.02% |
| | | **Ours** | **0.20** | **1.70** | **1.28** | **1.35** | **0.91** | **44.66%** |

**Table 6:** Comparison of the generated code efficiency on HUMANEVAL+ and COFFE. Methods that explicitly optimize efficiency are shaded; best results are in **bold**.

## D.2 Additional Results on Method Processing Time

To further support the findings presented in the main paper, we report the median processing time during the decoding phase for each method on the Mercury, HumanEval+, and COFFE datasets in Tables 7–9. These results serve as a supplement to Table 2 in the main paper.

Consistent with our earlier findings, our method demonstrates a significant advantage in decoding efficiency compared to baseline methods specifically designed to optimize the generated code beyond the perplexity, e.g., `Self-Debug`, `EffiLearner` and `PerfCodeGen`. These baselines achieve improved runtime performance or correctness in the generated outputs, but do so at the expense of substantially higher decoding latency—in some cases, one to two orders of magnitude slower than our method. In contrast, our approach achieves superior code efficiency, competitive or superior functional correctness, while maintaining fast generation speeds. It is worth noting that the vanilla baselines not explicitly optimized for code efficiency (i.e., Perplexity) remain faster, but they do not offer the same runtime benefits in the generated code as our approach.

| Models | Perplexity (Best-of-n) | Perplexity (Top-p) | Self-Debug | EffiLearner | PerfCodeGen | Ours |
|---|---|---|---|---|---|---|
| DeepSeek-6.7b | 2.89 | 2.22 | 3998.65 | 3555.88 | 3979.42 | 25.17 |
| OpenCoder-8b | 2.91 | 2.05 | 4117.52 | 4242.41 | 4884.11 | 23.36 |
| StarCoder2-15b | 6.06 | 5.13 | 7111.11 | 5989.32 | 7373.07 | 153.32 |
| DeepseekCoder-v2-16b | 6.27 | 5.84 | 7215.69 | 6004.49 | 7517.46 | 65.59 |
| Qwen2.5-Coder-32b | 13.33 | 11.03 | 25543.87 | 18787.55 | 20089.51 | 115.58 |

**Table 7:** The *median processing time* (in seconds) of each method on the MERCURY dataset.

| Models | Perplexity (Best-of-n) | Perplexity (Top-p) | Self-Debug | EffiLearner | PerfCodeGen | Ours |
|---|---|---|---|---|---|---|
| DeepSeek-6.7b | 3.18 | 2.45 | 4212.84 | 3763.19 | 4189.66 | 27.14 |
| OpenCoder-8b | 3.21 | 2.28 | 4328.75 | 4463.52 | 5078.94 | 25.36 |
| StarCoder2-15b | 6.61 | 5.52 | 7358.99 | 6224.08 | 7613.42 | 159.47 |
| DeepseekCoder-v2-16b | 6.68 | 6.21 | 7432.77 | 6230.55 | 7735.28 | 69.18 |
| Qwen2.5-Coder-32b | 14.02 | 11.59 | 26438.73 | 19420.86 | 20833.11 | 121.46 |

**Table 8:** The *median processing time* (in seconds) of each method on the HUMANEVAL+ dataset.

| Models | Perplexity (Best-of-n) | Perplexity (Top-p) | Self-Debug | EffiLearner | PerfCodeGen | Ours |
|---|---|---|---|---|---|---|
| DeepSeek-6.7b | 2.74 | 2.08 | 3895.42 | 3450.31 | 3870.15 | 24.38 |
| OpenCoder-8b | 2.78 | 1.95 | 4012.67 | 4129.84 | 4775.33 | 22.41 |
| StarCoder2-15b | 5.83 | 4.90 | 6982.76 | 5855.24 | 7232.58 | 149.11 |
| DeepseekCoder-v2-16b | 6.01 | 5.60 | 7099.28 | 5883.20 | 7400.38 | 63.45 |
| Qwen2.5-Coder-32b | 12.95 | 10.67 | 25032.15 | 18345.77 | 19675.28 | 112.76 |

**Table 9:** The *median processing time* (in seconds) of each method on the COFFE dataset.

## D.3 Hyperparameter Tuning

As part of our preliminary study, we performed hyperparameter tuning to determine effective configurations for sampling strategies in code generation.

This tuning was performed using the CodeLlama-7B-Instruct-HF and CodeLlama-13B-Instruct-HF models on the HumanEval+ dataset. The results of these experiments, detailing the average execution time (ET) of generated code under various hyperparameter settings, are presented in Table 10.

## D.4 Impact of Different Critique LLMs

We present the detailed quantitative results of varying critique LLMs in Table 11. This serves as a supplementary analysis to Table 3 in the main paper.

| $\alpha$ | $\beta$ | $\gamma$ | CodeLlama-7b-Instruct-HF | CodeLlama-13b-Instruct-HF |
|---|---|---|---|---|
| 1 | 1 | 0.5 | 1.61 | 0.21 |
| 1.3 | 1 | 0.5 | 1.55 | 0.19 |
| 1.3 | 1 | 0.4 | 1.54 | 0.17 |
| 1.5 | 1 | 0.5 | 1.49 | 0.19 |
| 1.5 | 0.7 | 0.3 | 1.56 | 0.18 |

**Table 10:** The ET (Avg) of the generated code on HUMANEVAL+ dataset across different settings.

| Critique LLMs | ET (Avg) | ET (Median) | NET (Avg) | NET (Median) | NMU | Correctness |
|---|---|---|---|---|---|---|
| DeepSeek-6.7b | 0.04 | 0.92 | 0.66 | 0.63 | 1.42 | 63.89% |
| CodeLlama-7b | 0.04 | 0.97 | 0.68 | 0.66 | 1.22 | 61.13% |
| OpenCoder-8b | 0.04 | 0.86 | 0.62 | 0.59 | 0.89 | 64.17% |
| DeepseekCoder-v2-16b | 0.04 | 0.85 | 0.61 | 0.59 | 0.92 | 64.79% |
| Qwen2.5-Coder-32b | 0.04 | 0.83 | 0.60 | 0.58 | 0.88 | 65.02% |

**Table 11:** More metrics on the impact of different critique LLMs of our method with OpenCoder-8b as the target generation LLM on EFFIBENCH.

## D.5 Impact of Intermediate Per-statement Selection

We present additional results comparing two variants: one that performs selection only at the end using standard perplexity-driven token generation during intermediate decoding (``End-Selection''), and our default setting that applies selection after every statement (``Ours''). The results in Table 12 show that our method consistently outperforms ``End-Selection'', suggesting that intermediate feedback more effectively guides generation and promotes greater diversity in the outputs. We hypothesize that applying the reward at the statement level (as the default of our method) provides fine-grained control during decoding, helping the model better explore its learned code manifold and construct efficient solutions incrementally, rather than relying solely on post hoc refinement.

| Models | Methods | ET(Avg) ↓ | Correctness ↑ |
|---|---|---|---|
| DeepSeek-6.7b | End-Selection | 0.43 | 33.79% |
| | Ours | 0.12 | 34.23% |
| OpenCoder-8b | End-Selection | 1.12 | 31.19% |
| | Ours | 0.13 | 32.88% |
| StarCoder2-15b-instruct | End-Selection | 0.78 | 33.15% |
| | Ours | 0.09 | 37.02% |
| DeepseekCoder-v2-16b | End-Selection | 1.08 | 30.03% |
| | Ours | 0.11 | 35.01% |
| Qwen2.5-33b | End-Selection | 0.79 | 30.08% |
| | Ours | 0.10 | 33.77% |

**Table 12:** Comparison between End-Selection and Ours across different models on MERCURY.

## D.6 Impact of Different Combinations of Configuration Components

We further investigate how different combinations of configuration components influence the efficiency of generated code across the COFFE and HUMANEVAL+ benchmarks. Figures 6 and 7 present the results. Notably, while individual components vary in their contributions, combining all different rewards generally yields the most effective outcomes in terms of code efficiency. Among the components, the AST reward appears to contribute the least when used in isolation.

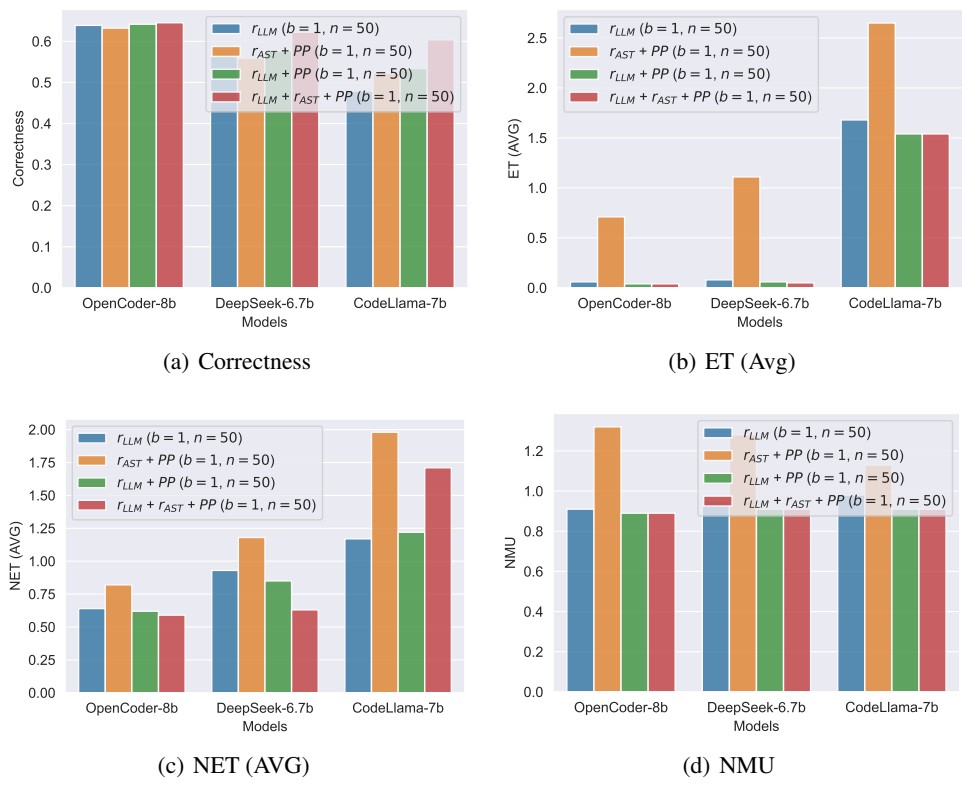

(a) Correctness

(b) ET (Avg)

(c) NET (AVG)

(d) NMU

**Figure 6:** Comparison across different configurations on HUMANEVAL+ with OpenCoder-8B as $f_\theta$.

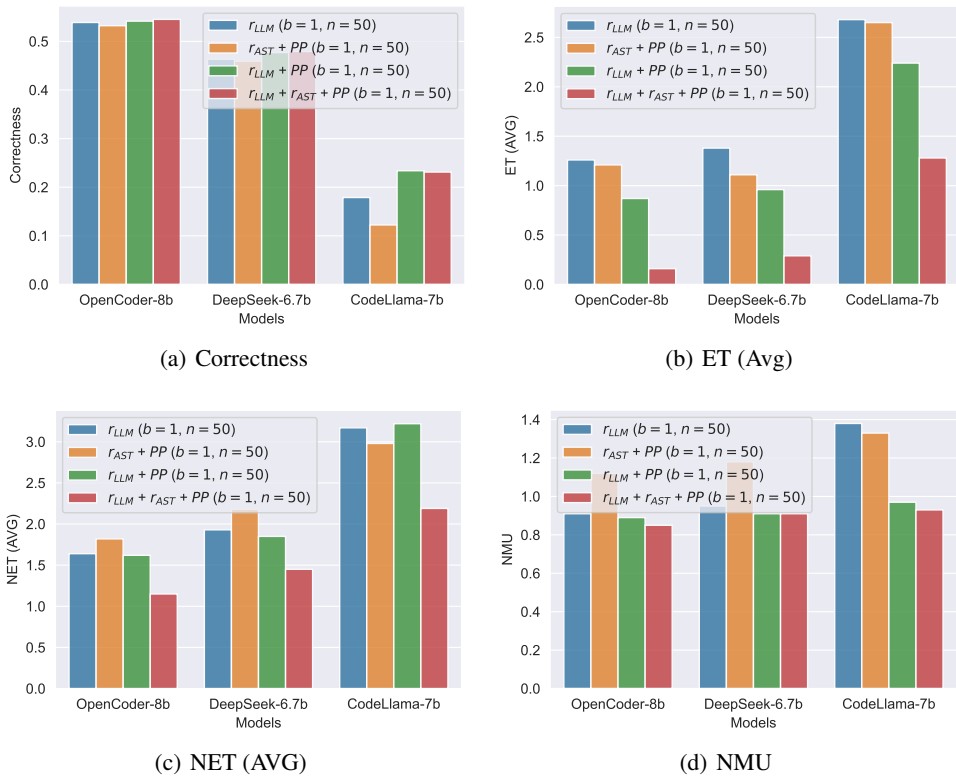

(a) Correctness

(b) ET (Avg)

(c) NET (AVG)

(d) NMU

**Figure 7:** Comparison across different configurations on COFFE with OpenCoder-8B as $f_\theta$.

