# OpenReview forum: "More Than Just Functional: LLM-as-a-Critique for Efficient Code Generation"
_NeurIPS.cc/2025/Conference — NeurIPS 2025 poster_

### Official Review · Reviewer_RQoh · 2025-06-11

**Clarity:** 2
**Significance:** 3
**Originality:** 3
**Rating:** 4
**Confidence:** 3

**Summary:**

This work introduces a novel approach for generating efficient code that leverages the LLM itself as an execution-free efficiency critique.
By prompting LLMs to analyze code static features like algorithmic complexity and AST structure, it generates an efficiency reward signal that navigates the decoding process toward code generations that are not only functionally correct but also computationally efficient.
Extensive experiments show that this approach significantly reduces execution time and memory usage across multiple benchmarks and models, outperforming costly execution-based refinement methods while improving functional correctness at the same time.

**Questions:**

- In Section Prompting LLMs as an Efficiency Critique, have you tried to gauge *runtime performance* and *memory usage efficiency* the real overhead? How well do the predicted values ​​match the actual values?

**Ethical Concerns:**

["NO or VERY MINOR ethics concerns only"]

**Final Justification:**

I thank the authors for their thorough rebuttal.

Their responses have addressed my main concerns, and I have finalized my score at 4.

**Limitations:**

Yes.

**Quality:**

2

**Strengths And Weaknesses:**

**Strengths**
- **Novel Approach:**
This work tackles code efficiency, a critical yet often overlooked problem in LLM code generation. It leverages LLM itself as a static effiency critique to bypass the costly code execution stage. This approach significantly reduces the inference overhead in generating efficient code comparing with existing approaches.
- **Comprehensive Empirical Evaluation:**
The authors conduct a thorough empirical study across multiple established benchmarks and a diverse set of state-of-the-art code LLMs. The proposed method consistently demonstrates a clear advantage over several efficient code generation baselines, achieving significant reductions in both execution time and memory usage.

**Weaknesses**
- **Dependence on Capable Critique LLMs:**
The method's performance is highly tied to the capabilities of the critique LLM as shown in Table 3. While predicting code performance without real execution is a promising direction, it remains unclear how well these predictions align with ground-truth performance metrics.

- **LLM Critique Generalization:**
Current evaluation is performed on self-contained and established benchmarks. It is highly probable that this data was involved in the training corpora of most LLMs. Therefore, it raises a concerns about whether the LLM's ability to critique code efficiency stems from truely code comprehension or merely from memorization of the benchmark solutions.

---

> ### Author Rebuttal · Authors · 2025-07-31
>
> We sincerely thank the reviewers for their constructive feedback! We are pleased that most reviewers have a clear positive view of our submission, praising it as a **"​​Novel Approach"** (Reviewer RQoh), of  **"great importance to industry community"** (Reviewer jQdC), is **"​​lightweight and easy to apply"** (Reviewer 4J1w), and **"presents a solid execution of a straightforward idea"** (Reviewer pDaa),  while also highlighting the **"thorough empirical study across multiple established benchmarks"** (Reviewer RQoh).
>
> We below address the individual concerns of **Reviewer RQoh**. All minor points will be directly incorporated into the revised manuscript.
>
> **Q1. [Dependence on Capable Critique LLMs: The method's performance is highly tied to the capabilities of the critique LLM as shown in Table 3. While predicting code performance without real execution is a promising direction, it remains unclear how well these predictions align with ground-truth performance metrics.]**
>
>
> First, while the capabilities of the critique LLM do influence performance, we emphasize that all standard evaluation results in our paper use the **same** model for both code generation and critique. This design ensures that our improvements do not rely on access to a stronger or external model, and thus impose no additional requirements.
>
> We provide supplementary empirical evidence showing that the LLM-based reward used during decoding aligns well with real execution efficiency. Specifically, we report the average Pearson correlation coefficients between the LLM reward scores and measured execution times across samples from multiple datasets for the studied models, achieving correlations of -0.78, -0.69, and -0.74 on Effibench, Mercury, and COFFE datasets, respectively. These results indicate a strong and consistent alignment between the LLM-generated reward signal and ground truth performance, supporting the reliability of our critique mechanism.
>
>
> **Q2. [LLM Critique Generalization: Current evaluation is performed on self-contained and established benchmarks. It is highly probable that this data was involved in the training corpora of most LLMs. Therefore, it raises a concerns about whether the LLM's ability to critique code efficiency stems from truly code comprehension or merely from memorization of the benchmark solutions.]**
>
> We acknowledge the concern regarding potential data contamination in benchmark evaluations, which is a challenge that is nearly impossible to eliminate entirely, given that LLMs are typically trained on large-scale web-scraped corpora. However, our results suggest that even when LLMs possess the necessary knowledge to generate correct and efficient code, they often fail to do so. This is likely due to limitations in autoregressive decoding and the reliance on perplexity as the primary generation objective. In this context, our approach of steering the LLM at inference time using feedback signals (derived from the *same* model) should offer meaningful practical value by promoting generation that better aligns with both correctness and efficiency.

---

> ### Comment · Reviewer_RQoh · 2025-08-05
>
> Thank you for the clarifications. Could you elaborate on how you calculate the Pearson correlation coefficients? I'm also curious if you used an isolated environment for code execution and whether the results would be consistent if run on different machines.

---

> > ### Author Response · Authors · 2025-08-05
> >
> > We sincerely thank you again for taking the time to review our paper and for your valuable and insightful feedback. Below, we provide additional clarification on the feedback.
> >
> > **Q1. [Could you elaborate on how you calculate the Pearson correlation coefficients?]**
> > To compute the Pearson correlation coefficients, we extract the LLM reward score for the final code sample generated by our approach. We then measure its execution time and subtract the execution time of the corresponding baseline, i.e., “perplexity (best-of-n)”. This difference captures the relative efficiency changes and helps eliminate the effect of the inherent difficulty of individual programs. The Pearson correlation coefficient is then calculated between the LLM reward scores and these execution time differences to evaluate how well the reward signal aligns with actual performance gains.
> >
> > **Q2. [I'm also curious if you used an isolated environment for code execution and whether the results would be consistent if run on different machines.]**
> >
> > All execution time measurements were conducted in isolated environments. Each test code was executed in a separate virtual machine instance with identical configurations to minimize system-level variance. While execution times may vary slightly across different hardware setups, our findings remain consistent as long as all evaluations are performed within the same controlled environment.

---

> > > ### Comment · Reviewer_RQoh · 2025-08-07
> > >
> > > Thank you for taking time to address my concerns. I'm positive on this paper and I will maintain my score for acceptance.

---

> > > > ### Author Response · Authors · 2025-08-08
> > > >
> > > > We are pleased that our rebuttal has addressed your concerns, and we thank you again for your valuable feedback. We will incorporate them into our revision.

---

### Official Review · Reviewer_pDaa · 2025-06-30

**Clarity:** 3
**Significance:** 3
**Originality:** 3
**Rating:** 5
**Confidence:** 3

**Summary:**

The paper proposes a method for better utilizing the existing capabilities of code-generating LLMs to produce more efficient code. The two (2) key ideas in this paper are to (1) introduce an efficiency-guided critique as an extension to the code-generation reward function and (2) to make that reward function a linear combination of directly measurable static AST features and LLM-generated estimates of other performance metrics, including time and space complexity, runtime performance and memory efficiency. The paper experiments with various configurations of the reward function, including both direct and LLM measurement of the AST analysis. Rewards are evaluated at the statement level and generation is organized using a beam search method. The approach has the advantages of improving generated code efficiency without requiring model retraining (or the associated augmented datasets required for retraining) and not requiring runtime evaluation (or standing up the environments needed for runtime evaluation). Since the approach does not involve retraining or code execution, it generalizes across existing codegen models. The paper claims a 17x average execution speedup vs SotA results (Effilearner, PerfCodeGen) measured using EffiBench, HumanEval+, Mercury and COFFE against exiting code generation models, including DeepSeekCoder, OpenCoder, StarCoder and Qwen-2.5-Coder.

**Questions:**

Questions are contained within the strengths and weaknesses section of the review.

It summary, the paper demonstrates a strong result from a straightforward idea of combining static and LLM-as-judge metrics into a reward function. As a research paper, I would like to come away with a better understanding of where the performance gains are coming from. This could take the form of any or all of (1) more quantitative information that shows the impact of various design choices (varying alpha/beta, comparison with post-generation LLM-driven refactoring), (2) some additional examples that show why the method finds performance gains that a system like Effi-Learner misses, or (3) some analyis on the accuracy of the LLM-provided estimates used to drive the reward function and if the accuracy of these estimates correlate with reported gains.

The result is solid and I would be open to improving the rating if the weaknesses can be addressed, even as an addition to the Appendix.

**Ethical Concerns:**

["NO or VERY MINOR ethics concerns only"]

**Final Justification:**

I appreciated the author's rebuttal and their addressing my feedback with additional results. I have also looked at the material provided as response to other reviewer's comments. On the basis of these clarifications and improvements I have increased my overall rating to 5.

**Limitations:**

The paper includes a section of limitations. The premise that "a trained LLM already possesses the ability to understand code efficiency" seems like something that could have been measured by checking how accurate the various critique estimates actually are. The limittations section does not comment on likely applicablity to larger code bases or other languages.

**Paper Formatting Concerns:**

No formatting concerns.

**Quality:**

3

**Strengths And Weaknesses:**

Strength - The paper presents a solid execution of a straightforward idea and demonstrates how it leads to an improvement on basic coding benchmarks. Prompts and AST metrics used and other details needed to support reproducibility are provided in the appendix.

Strength - The approach is practical at least in so far as it does not require additional training and does not require setting up a runtime environment or sandbox for code execution.

Strength - The paper leverages existing code benchmarks and tests a reasonable selection of code generation models as described in the summary. Comparisons against recent code-efficiency improvements systems, including Effi-Learner, are good.

Strength/Weakness - The related work section is adequate, but I would have expected more discussion of refactoring systems. That would seem to the biggest comparable to this work - systems that reduce the complexity of working code after the solution is fully formed, but still without requiring code execution. What was the advantage of baking code-efficiency into the reward function and applying it at the statement level?

Weakness - While example outcomes are provided in Figures 2-5, it would be helpful to have more information about what is driving the improvement in the method. While the paper provides a few different prompting strategies, I did not see a breakdown of the impact of varying the alpha and beta parameters in the general formulation. Is having the secondary LLM consider AST factors redundant with the empirical AST measure? There is some description of these tradeoffs beginning on line 291, but it would be helpful to understand in more detail what the tradeoff space looks like.

Weakness - The paper lacks much introspection into understanding why the method is more effective than alternatives. Why is correctness improved? Is shorter code generally more likely to be correct? What is the advantage of performing multi-dimensional analysis after every statement? How well do the individual estimates made on predicted performance compare to the actual results? While there is some commentary starting around line 250, there is only a minimal attempt at explaining why the approach achieves better results.

Weakness - There is essentially no commentary on how the method might generalize to other programming languages or larger code problems.

---

> ### Author Rebuttal · Authors · 2025-07-31
>
> We sincerely thank the reviewers for their constructive feedback! We are pleased that most reviewers have a clear positive view of our submission, praising it as a **"​​Novel Approach"** (Reviewer RQoh), of  **"great importance to industry community"** (Reviewer jQdC), is **"​​lightweight and easy to apply"** (Reviewer 4J1w), and **"presents a solid execution of a straightforward idea"** (Reviewer pDaa),  while also highlighting the **"thorough empirical study across multiple established benchmarks"** (Reviewer RQoh).
>
>
> We below address the individual concerns of **Reviewer pDaa**. All minor points will be directly incorporated into the revised manuscript.
>
> **Q1. [The related work section is adequate, but I would have expected more discussion of refactoring systems. What was the advantage of baking code-efficiency into the reward function and applying it at the statement level?]**
>
> We thank the reviewer for highlighting this point. We agree that automatic refactoring systems are worth discussing. Traditional refactoring tools (such as those in IDEs or static analyzers) aim to improve code quality or structure while preserving behavior. While recent work has explored learning-based approaches, most existing methods remain largely template-driven and rely on executability and test cases to validate behavior-preserving transformations. As a result, they often focus on stylistic or structural improvements and are less effective at producing efficiency-oriented rewrites, particularly without access to execution or profiling.
>
> In contrast, our approach incorporates efficiency considerations directly into the sampling objective through an inference-time reward function. By applying this at the statement level, it enables fine-grained control over decoding, helping the model explore its learned code manifold and generate efficient solutions as part of the construction process, rather than just refactoring them post hoc.
>
> We also believe that insights from refactoring systems could inform future improvements to our refinement stage, and we will revise the related work section to reflect this connection more clearly.
>
> **Q2. [Breakdown of the impact of varying the alpha and beta parameters in the general formulation. Is having the secondary LLM consider AST factors redundant with the empirical AST measure?]**
>
> We present detailed results on hyperparameter tuning using CodeLlama-7B-Instruct on the HumanEval+ dataset. While it is challenging to draw definitive trends given the interplay of three correlated reward components, we generally observe that incorporating all terms leads to the best overall performance. Among them, slightly emphasizing the LLM-based reward ($\beta$) over the perplexity-based term ($\gamma$), e.g., using a weight ratio of 3:2 to 2:1 tends to yield better results. Additionally, modestly increasing the weight of the AST term ($\alpha$) also leads to slight improvements in final performance.
>
> Regarding the AST term, our design leverages AST signals in two complementary ways. First, they provide knowledge guidance to the LLM reward model via prompting, as incorporating AST information helps the LLM reason about structural aspects of efficiency—something we find empirically beneficial. Second, because AST alignment is closely tied to structural correctness, where deviations may compromise functionality, it serves a role akin to a *hard* constraint. This contrasts with the LLM-based critique, which reflects a *soft*, semantic preference. This separation motivates us to modulate their respective influences via the $\alpha$ and $\beta$ parameters, ensuring that structural correctness is explicitly prioritized alongside more flexible semantic signals.
>
>
> |  $\alpha$  |  $\beta$  |  $\gamma$  | ET (Avg) | Correctness |
> |:---:|:---:|:---:|:-----------:|:-----------:|
> |  1  |  0  |  0  |     4.89    |    38.27%   |
> |  0  |  1  |  0  |     1.68    |    50.24%   |
> |  0  |  0  |  1  |     2.78    |    45.43%   |
> |  0  |  1  |  1  |     1.67    |    52.55%   |
> | 0.5 |  1  |  1  |     1.64    |    54.49%   |
> |  1  | 0.5 |  1  |     1.72    |    52.25%   |
> |  1  |  1  | 0.5 |     1.61    |    56.63%   |
> |  1  |  1  |  1  |     1.65    |    49.95%   |
> |  1  |  1  | 1.5 |     1.71    |    47.78%   |
> |  1  | 1.5 |  1  |     1.57    |    53.38%   |
> |  1  |  2  |  1  |     1.58    |    55.55%   |
> | 1.5 |  1  |  1  |     1.63    |    51.13%   |
> | 1.4 |  1  | 0.5 |     1.59    |    57.77%   |
> | 1.4 |  1  | 0.4 |     1.62    |    55.58%   |
> | 1.3 |  1  | 0.5 |     1.55    |    59.12%   |
> | 1.3 |  1  | 0.4 |     1.54    |    60.37%   |
> | 1.3 |  1  | 0.3 |     1.57    |    59.08%   |
> | 1.3 |  1  | 0.2 |     1.62    |    58.88%   |
> | 1.3 | 0.7 | 0.4 |     1.62    |    51.21%   |
>
>
> **Q3. [Understanding why the method is more effective than alternatives. Why is correctness improved? Why the method find performance gains that a system like Effi-Learner misses?]**
>
> We believe that incorporating AST patterns into the decoding-time reward plays an important role in guiding the model to be aware of both correctness and efficiency during generation. In contrast, previous refinement-based methods typically rely on passively re-feeding profiling signals and simply filtering out incorrect code that fails test cases for further refinement. However, this process lacks active guidance and can lead to instability—new errors may emerge when attempting to fix earlier ones, as seen in approaches like Self-Debug and Self-Refinement. Without an explicit feedback signal guiding the generation holistically, these methods risk oscillating between incorrect variants rather than converging on efficient and correct solutions.
>
>
> **Q4. [Is shorter code generally more likely to be correct? What is the advantage of performing multi-dimensional analysis after every statement?]**
>
> Regarding code length: We compute the Pearson correlation between average code length (in lines) and correctness across our experiments and find a coefficient of 0.06, indicating a negligible relationship between the two. The observed code lengths are likely an artifact of our example selection for visualization purposes, as we deliberately chose shorter examples to fit within space constraints.
>
>
> Regarding the advantage of performing analysis *after every statement*: We present additional results comparing two variants: one that performs selection only at the end using standard perplexity-driven token generation during intermediate decoding (**End-Selection**), and our default setting that applies selection after every statement (**Ours**). The results show that our method consistently outperforms End-Selection, suggesting that intermediate feedback more effectively guides generation and promotes greater diversity in the outputs. As briefly mentioned above, we hypothesize that applying the reward at the statement level provides fine-grained control during decoding, helping the model better explore its learned code manifold and construct efficient solutions incrementally, rather than relying solely on post hoc refinement.
>
>
>
>
> |Models|Methods|ET(AVG)|Correctness|
> |:-----|:------|:------:|:----------:|
> |DeepSeek-6.7b|End-Selection|0.43|33.79%|
> ||Ours|0.12|34.23%|
> |OpenCoder-8b|End-Selection|1.12|31.19%|
> ||Ours|0.13|32.88%|
> |startcoder2-15b-instruct|End-Selection|0.78|33.15%|
> ||Ours|0.09|37.02%|
> |deepseekcoder-v2-16B|End-Selection|1.08|30.03%|
> ||Ours|0.11|35.01%|
> |qwen2.5-33b|End-Selection|0.79|30.08%|
> ||Ours|0.10|33.77%|
>
> **Q5. [How well do the individual estimates made on predicted performance compare to the actual results?]**
>
> We provide supplementary empirical evidence showing that the LLM-based reward used during decoding aligns well with real execution efficiency. Specifically, we report the average Pearson correlation coefficients between the LLM reward scores and measured execution times across samples from multiple datasets for the studied models, achieving correlations of -0.78, -0.69, and -0.74 on Effibench, Mercury, and COFFE datasets, respectively. These results indeed indicate a strong and consistent alignment between the LLM-generated reward signal and ground truth performance.
>
>
> **Q6. [There is essentially no commentary on how the method might generalize to other programming languages or larger code problems.]**
>
> We appreciate the reviewer’s comment and will include a discussion on generalization in the revised version. To our knowledge, there is no fundamental limitation that prevents our method from extending to other programming languages. High-level AST patterns are typically identical or highly similar across languages, and the LLM-based reward component is designed to be language-agnostic and should remain effective, assuming the LLM is appropriately trained. While the AST extraction module would need to be adapted for each target language, this is a relatively straightforward engineering task. Additionally, the processing time of our method scales linearly with code length and, in principle, should not form a bottleneck for larger code problems.

---

> > ### Comment · Reviewer_pDaa · 2025-08-05
> >
> > Thank you for your rebuttal and for addressing all of my feedback. On Q1, there have been studies of LLM-based refactoring systems that do not rely on templates or execution. This is where I was hoping to see some comparison in the related work. The responses to Q2-Q5 provide a lot of useful information that it would be great to see in the appendix - thank you for putting this together. On Q6, it is entirely reasonable that the paper is restricted to Python, but the commentary on how it might translate to other languages is interesting. Your thorough and thoughtful responses are appreciated.

---

> > > ### Author Response · Authors · 2025-08-05
> > >
> > > We sincerely thank you again for taking the time to review our paper and for your valuable and insightful feedback. We appreciate your constructive comments and are glad you found our clarifications helpful. We will incorporate clarifications addressing all the feedback in the revision. Below, we provide additional discussion addressing Q1 and Q6.
> > >
> > >
> > > **Q1: [There have been studies of LLM-based refactoring systems that do not rely on templates or execution. This is where I was hoping to see some comparison in the related work.]**
> > >
> > > While the recent LLM-based refactoring systems represent a shift beyond traditional rule-based approaches, they remain focused on improving internal code quality (i.e., enhancing readability, modularity, and structural maintainability) [1], typically measured through metrics like coupling, cohesion, modularity, and cyclomatic complexity [2,3]. In contrast, our work focuses on downstream execution efficiency, which LLM-driven refactoring systems are not designed for.
> > > Broadly, our focus on execution efficiency can complement traditional refactoring by addressing aspects of code quality that extend beyond structure or style, particularly in performance-critical contexts. Conversely, practical insights from refactoring systems may also enhance our code generation approach. We will clarify this distinction more explicitly in the related work section and acknowledge that a deeper comparative analysis could be a valuable direction for future research.
> > >
> > >
> > > [1] Cordeiro et al. "LLM-Driven Code Refactoring: Opportunities and Limitations", IEEE/ACM Second IDE Workshop 2025.
> > > [2] DePalma et al. "Exploring ChatGPT’s code refactoring capabilities: An empirical study", Expert Systems with Applications 2024.
> > > [3] Cordeiro et al. "An empirical study on the code refactoring capability of large language models", Arxiv 2024.
> > >
> > > **Q6. [It is entirely reasonable that the paper is restricted to Python, but the commentary on how it might translate to other languages is interesting]**
> > >
> > > We additionally conducted a preliminary experiment in Java using the Java portion of the HumanEval-X dataset, which includes 164 programs. Our approach continues to show performance gains when evaluated with the DeepSeek-R1-Distill-Qwen-32B model.  Since the other baselines were originally implemented for Python and would require substantial reimplementation to run on this dataset, they are omitted from the comparison below.
> > >
> > > | Methods | ET(AVG) | NMU | Time Cost (Median) | Correctness |
> > > |:---:|:---:|:---:|:---:|:---:|
> > > | Default(temperature=0.6, top-p=0.95) | 0.0069 | 1.17 | 15.74 | 14.47% |
> > > | Original (Best-of-n, n=50) | 0.0057 | 1.15 | 23.95 | 13.39% |
> > > | Original (Top P) | 0.0066 | 1.22 | 14.98 | 14.21% |
> > > | Self-Debug | 0.0065 | 0.93 | 24452.38 | 28.73% |
> > > | Ours | 0.0028 | 0.88 | 189.06 | 35.65% |

---

> > > > ### Author Response · Authors · 2025-08-08
> > > >
> > > > We sincerely thank you again for your insightful and constructive feedback, as well as your support for our work. We will incorporate your comments into the revision.

---

### Official Review · Reviewer_4J1w · 2025-07-01

**Clarity:** 3
**Significance:** 2
**Originality:** 3
**Rating:** 4
**Confidence:** 4

**Summary:**

Based on my understanding, this paper introduces a clever and surprisingly practical approach to improving the efficiency of code generated by large language models (LLMs). While most prior work and current coding assistants focus mainly on ensuring the code is functional and syntactically correct, this work points out a critical yet often overlooked problem: LLM-generated code can be very inefficient, with longer runtime and higher memory usage compared to human-written code. The interesting point is that LLMs actually do have implicit knowledge about what makes code efficient, but standard training objectives (like perplexity) and decoding strategies fail to activate this knowledge. So, the authors propose a simple solution, using a second LLM, strategically prompted, as an "efficiency critic" to guide the code generation process. This efficiency-aware critique is based on the structure of the code (like the abstract syntax tree), and it doesn't require any actual code execution or expensive annotations, which making it  lightweight and easy to apply. (However, the authors proposed "time complexity, space complexity,
runtime performance, memory usage efficiency, syntax correctness" could be easily assessed by simple **speed test scripts**, I really not sure the meaning why introduce the LLM to do this.)

**Questions:**

The authors propose that "time complexity, space complexity, runtime performance, memory usage efficiency, and syntax correctness" can be assessed by simple speed test scripts. I am not convinced of the rationale behind introducing an LLM to perform this evaluation. Others see weaknesses. I am open to discussion about this key point.

**Ethical Concerns:**

["NO or VERY MINOR ethics concerns only"]

**Final Justification:**

The core assumption for this task is not convinced.

**Limitations:**

yes

**Quality:**

2

**Strengths And Weaknesses:**

Strength:

The paper reveals that LLMs already possess implicit knowledge about code efficiency (e.g., runtime/memory complexity), even though they are typically trained for correctness. The method taps into this latent knowledge without modifying the model. Instead of retraining or fine-tuning the LLM, the method simply prompts a secondary LLM to act as an efficiency critic.

Weaknesses:

1. While using LLMs as critics or evaluators is an emerging trend, applying them to assess fine-grained, precision-sensitive attributes (e.g., efficiency, memory use) raises concerns. The paper assumes LLMs can act as reliable efficiency evaluators, but this assumption lacks empirical validation or calibration against ground truth measurements. Unlike prior works that rigorously validate LLM-as-judger setups, this paper does not justify the trustworthiness of its critique model with sufficient rigor.

2. The proposed method hinges on the assumption that LLMs can consistently and accurately judge code efficiency without actual execution. However, inspection of examples and appendix results reveals that the judgments may be inconsistent or even contradictory across similar samples. For a critique system to guide generation, its correctness must be established—something not thoroughly validated here.

3. The decoding configuration described in lines 212–214 (“best-of-n=50 with top-p=0.95 sampling”) is claimed to follow [33], but [33] in fact uses different sampling parameters. Moreover, the search space size (n=50) is unusually large and can significantly impact performance, especially under resource constraints. This discrepancy is not analyzed or ablated, making it difficult to disentangle gains from prompt-based critique versus aggressive sampling.

---

> ### Author Rebuttal · Authors · 2025-07-31
>
> We sincerely thank the reviewers for their constructive feedback! We are pleased that most reviewers have a clear positive view of our submission, praising it as a **"​​Novel Approach"** (Reviewer RQoh), of  **"great importance to industry community"** (Reviewer jQdC), is **"​​lightweight and easy to apply"** (Reviewer 4J1w), and **"presents a solid execution of a straightforward idea"** (Reviewer pDaa),  while also highlighting the **"thorough empirical study across multiple established benchmarks"** (Reviewer RQoh).
>
> We below address the individual concerns of **Reviewer 4J1w**. All minor points will be directly incorporated into the revised manuscript.
>
> **Q1. [While using LLMs as critics or evaluators is an emerging trend, applying them to assess fine-grained, precision-sensitive attributes (e.g., efficiency, memory use) raises concerns. The paper assumes LLMs can act as reliable efficiency evaluators, but this assumption lacks empirical validation or calibration against ground truth measurements. Unlike prior works that rigorously validate LLM-as-judger setups, this paper does not justify the trustworthiness of its critique model with sufficient rigor.]**
>
> First, we would like to clarify that our approach does **not** rely on LLMs as the final evaluators for measuring code efficiency. All evaluation results reported in the paper are based on *real code execution*, ensuring objectivity, verifiability, and immunity to potential biases inherent in LLM-as-judge evaluations. Therefore, our findings are grounded in actual runtime behavior, not subjective estimations.
>
> In addition, we provide supplementary empirical evidence showing that the LLM-based reward used during decoding aligns well with real execution efficiency. Specifically, we report the average Pearson correlation coefficients between the LLM reward scores and measured execution times across samples from multiple datasets for the studied models, achieving correlations of -0.78, -0.69, and -0.74 on Effibench, Mercury, and COFFE datasets, respectively. These results indeed indicate a strong and consistent alignment between the LLM-generated reward signal and ground truth performance.
>
>
>
> **Q2. [The proposed method hinges on the assumption that LLMs can consistently and accurately judge code efficiency without actual execution. However, inspection of examples and appendix results reveals that the judgments may be inconsistent or even contradictory across similar samples. For a critique system to guide generation, its correctness must be established—something not thoroughly validated here.]**
>
> We appreciate the reviewer’s concern and would like to clarify that the examples provided in Figures 2–5 actually demonstrate a consistent alignment between the LLM reward signal and real code execution performance. Specifically, higher LLM rewards (0.89, 0.81, 0.78, 0.43) closely correspond to lower execution times (1.64E-4, 6.45E-4, 5.97E-4, 1.52E-3), indicating the LLM’s ability to reasonably infer code efficiency. The slight variance between some scores (e.g., 0.81 vs. 0.78) is expected, as the LLM reward captures a combination of both time and space complexity, which may not align perfectly with a single metric like execution time.
>
> To our knowledge, Figures 2–5 are the only examples in the paper that include both LLM rewards and real execution metrics, and we are not aware of any other results that could suggest inconsistency or contradiction.
>
> Furthermore, as elaborated in our response to Q1, we provide supplementary empirical validation showing that the LLM reward signal indeed strongly correlates with ground-truth execution time across multiple datasets.
>
> **Q3. [The decoding configuration described in lines 212–214 (“best-of-n=50 with top-p=0.95 sampling”) is claimed to follow [33], but [33] in fact uses different sampling parameters. Moreover, the search space size (n=50) is unusually large and can significantly impact performance, especially under resource constraints. This discrepancy is not analyzed or ablated, making it difficult to disentangle gains from prompt-based critique versus aggressive sampling.]**
> In [33], the authors indeed claim to use top-p = 0.95 in Section 3 (page 4). In our experiments, we set best-of-n = 50 to ensure a fairer comparison and to isolate the impact of our proposed method from potential gains that could result simply from using a large n.
>
> Below, we present results on Qwen2.5-Coder-33B using our approach under varying values of n. The results show that increasing n leads to improved generated code efficiency and accuracy, albeit with higher computational cost. Overall, we choose n = 50 as our default setting, as it provides a strong balance between generation performance and computational efficiency.
>
> |  n | ET(AVG) | ET(Median) | NET(Avg) | NET(Median) |  NMU | Time Cost (Median) | Correctness |
> |:--:|:-------:|:----------:|:--------:|:-----------:|:----:|:------------------:|:-----------:|
> |  1 |   1.38  |    1.34    |   1.11   |     0.94    | 1.35 |        8.79        |    55.33%   |
> |  5 |   1.23  |    1.31    |   1.24   |     0.94    | 1.27 |        17.65       |    55.58%   |
> | 10 |   0.59  |    0.98    |   1.01   |     1.18    | 0.95 |        23.33       |    57.52%   |
> | 25 |   0.08  |    0.87    |   0.82   |     1.12    | 0.95 |        59.43       |    59.91%   |
> | 50 |   0.05  |    0.61    |   0.66   |     0.58    | 0.89 |       101.34       |    66.42%   |
> | 70 |   0.05  |    0.59    |   0.65   |     0.57    | 0.91 |       185.58       |    67.26%   |
>
>
> **Q4. [The authors propose that "time complexity, space complexity, runtime performance, memory usage efficiency, and syntax correctness" can be assessed by simple speed test scripts. I am not convinced of the rationale behind introducing an LLM to perform this evaluation.]**
>
> First, as noted above, we do *not* use the LLM for *final evaluation*. All final assessments are performed via actual code execution, which is straightforward and typically requires only one or a few runs per sample. In contrast, the LLM is used during the *refinement stage*, which often involves **repeated** sampling and evaluation to guide the model toward generating correct and efficient code. Replacing such **repeated** ground-truth executions in this stage with a lightweight LLM-based proxy results in substantial computational savings.
>
>
> Additionally, the human effort and infrastructure overhead involved in setting up execution environments are far from negligible. In this regard, our method presents a practical and scalable alternative “for free”: it enables iterative improvement toward efficient and correct code using only the **same** LLM, without the need for repeatedly executing the code. We believe this makes our approach particularly valuable in resource-constrained or rapid-development settings.

---

> > ### Author Response · Authors · 2025-08-05
> >
> > We sincerely thank you again for taking the time to review our paper and for your valuable and insightful feedback. We would like to kindly follow up to ask whether our rebuttal has addressed your concerns, and if you have any further questions or comments regarding our work. We are happy to provide additional clarification on any remaining issues if needed.

---

> ### Comment · Reviewer_4J1w · 2025-08-06
> **Thank you for the detailed rebuttal.**
>
> Thank you for the detailed rebuttal.
>
> While I appreciate the clarification that all final evaluations are based on real execution, my concern still lies with the motivation and reliability of using LLMs as efficiency critiques during decoding (I know the grounded in actual runtime behavior). Recent works [1, 2] (not only two works) have challenged LLM-as-judge approaches for fine-grained tasks which covers the metric in this paper, and which are often hardware-sensitive and not easily inferred from code area alone. Correlation with execution time is helpful but not sufficient to ensure trustworthiness without stronger calibration or adversarial validation. While the reported correlation scores are moderately strong, Pearson correlation alone is a necessary but insufficient condition to claim reliable alignment. It does not guarantee local consistency, accurate ranking, or robustness under distribution shifts. For a reward signal to guide decoding effectively, stronger calibration and per-sample analysis would be necessary. I believe the core assumption that LLMs can reliably guide efficiency without execution remains under-justified and weakens the motivation of the proposed approach.
>
> As the these concerns, I maintain the original score.
>
> [1] Preference Leakage: A Contamination Problem in LLM-as-a-judge
>
> [2] Adversarial ML Problems Are Getting Harder to Solve and to Evaluate

---

> > ### Author Response · Authors · 2025-08-06
> >
> > First of all, as the reviewer has also acknowledged, we would like to emphasize that our reported results are **“grounded in actual runtime behavior”**, which provides a truly **objective** basis for evaluation and is not subject to potential “preference bias” associated with LLM judging.
> >
> > While actual runtime performance is hardware-sensitive, we carefully controlled the evaluation environment to ensure reproducibility. Specifically, each test code was executed in an **isolated virtual machine instance with identical system configurations**, minimizing system-level variance. This setup aligns with best practices in the code generation community and represents one of the most reliable and reproducible standards for performance evaluation.
> >
> > Secondly, the **only metric we identified** in the referenced work is the so-called *“preference leakage score”*, which is unrelated to the code execution metric used in our study. As such, it is **unclear how the claim that these works "covers the metric in this paper" is substantiated**. As we have already noted, LLM-as-a-judge approaches are typically applied in domains where objective evaluation is difficult or impossible. In contrast, our setting involves direct, objective measurement, making our case fundamentally different. To the best of our knowledge, there are **no widely accepted use cases involving objectively measurable outcomes (such as ours) that are still regarded as untrustworthy** under controlled evaluation.
> >
> > More importantly, while we acknowledge concerns about the general capabilities of LLMs in open-domain tasks, the coding domain may differ in certain ways. The training corpus often contains **recurring (and potentially finite) efficiency-related (anti-)patterns** that models can learn. Although final runtime is influenced by hardware, the primary determinant of efficiency is the program’s abstract structure, as reflected in its underlying computational graph. **These structural properties are hardware-independent** and can be analyzed to assess code efficiency.
> >
> > Moreover, **efficiency-related information about specific code snippets is likely available in the LLM’s (pre-)training data**, as many online sources (such as websites and blogs) include discussions of runtime or algorithmic complexity alongside code. Prior work has also shown that machine learning models can, to some extent, predict runtime or performance given the programming code [1]. In addition, our final reward function **explicitly incorporates static analysis of the code’s abstract syntax tree (AST)**, which provides reliable and interpretable structural information to further support efficiency estimation.
> >
> > More broadly, the question of **“how accurate a critique must be”** remains open. Self-distillation and weak supervision have repeatedly been shown to be effective, even when the supervisory signal is imperfect. In fact, weak teachers/critiques can sometimes help train strong students when used strategically, as in methods like boosting. All these suggest that even imperfect signals could still offer useful guidance in the right context.

---

> ### Comment · Reviewer_4J1w · 2025-08-07
> **Response to Authors**
>
> Thank you for the detailed rebuttal. Most of my concerns have been addressed.
>
>
> **I have raised my score to weak acc. Wish the authors good luck!**

---

> > ### Author Response · Authors · 2025-08-08
> >
> > We sincerely appreciate your recognition and support of our work. We will incorporate your valuable suggestions into our revision.

---

### Official Review · Reviewer_jQdC · 2025-07-03

**Clarity:** 3
**Significance:** 2
**Originality:** 2
**Rating:** 4
**Confidence:** 3

**Summary:**

The paper targets the challenging problem of efficient code generation by presenting a new method. The main motivation is to integrate the LLM during the decoding processing as a critique for code efficiency. Experimental results on several benchmarks like EffiBench, HumanEval+, COFFE, and Mercury, as well as different coding models validate the effectiveness and efficiency of the proposed algorithm.

**Questions:**

Please mainly address the questions in the weakness section, especially, the questions related with the novelty of the paper.

**Ethical Concerns:**

["NO or VERY MINOR ethics concerns only"]

**Final Justification:**

The rebuttal well addressed most of my concerns in the previous round of review. Considering the simple design of the proposed approach as well as the extensive evaluations, I would suggest to borderline accept the paper.

**Limitations:**

The paper addressed the limitations in the paper.

**Quality:**

3

**Strengths And Weaknesses:**

# Strength
- The problem of efficient code generation is of great importance to the industry community. Thus,                        the improvement of the code generation efficiency should be repreciated.
- The design of the proposed algorithm is simple to implement. During the decoding process, it introduces several reward functions like AST pattern matching, LLM-prompting, and perplexity as the critique to improve code efficiency.
- Extensive experimental results have been evaluated on several benchmarks as well as different coding models. As shown in Table 1, the proposed algorithm obtains superior performance gain over the baseline.


# Weakness
- The novelty of the paper is limited. It introduces a set of reward functions as the critique to speed up the code efficiency. However, as it is only conducted on the decoding stage, it cannot boost the performance of the original coding model. If the coding model itself cannot generate both effecient and effective codes, the proposed algorithm will fail as well.
- The proposed algorithm will obviously increase the decoding time of the coding model. How about a comparison with the COT/RL model like Deepseek R1 for both inference computational cost as well as the code efficiency.
- How about the setting of the hyperparameters (\alpha, \beta, \gamma) in General Formulation reward signals (there should have an equation number in the paper)? Is there any ablation study to analyse the setting of these hyperparameters?

---

> ### Author Rebuttal · Authors · 2025-07-31
>
> We sincerely thank the reviewers for their constructive feedback! We are pleased that most reviewers have a clear positive view of our submission, praising it as a **"​​Novel Approach"** (Reviewer RQoh), of  **"great importance to industry community"** (Reviewer jQdC), is **"​​lightweight and easy to apply"** (Reviewer 4J1w), and **"presents a solid execution of a straightforward idea"** (Reviewer pDaa),  while also highlighting the **"thorough empirical study across multiple established benchmarks"** (Reviewer RQoh).
>
> We below address the individual concerns of **Reviewer jQdC**. All minor points will be directly incorporated into the revised manuscript.
>
> **Q1. [The novelty of the paper is limited. It introduces a set of reward functions as the critique to speed up the code efficiency. However, as it is only conducted on the decoding stage, it cannot boost the performance of the original coding model. If the coding model itself cannot generate both efficient and effective codes, the proposed algorithm will fail as well.]**
>
> To our knowledge, we are the first to move away from the dominant paradigm of relying on labor- and time-intensive real execution for refining code generation, and instead propose an alternative that uses a set of lightweight, efficiency-oriented reward functions directly during decoding.
>
> While it is difficult, if not impossible, to make definitive claims about the inherent capabilities of a code generation model, what we do observe is that even when a model has the capacity to produce correct and efficient code, it often fails to do so (likely due to the nature of autoregressive generation being guided by token-level perplexity rather than functional objectives). Our method effectively steers the generation process toward more efficient and reliable outputs without modifying the base model or relying on ground-truth execution. We believe this shift in methodology offers substantial practical value, particularly in scenarios where real-time execution feedback is costly or infeasible.
>
>
> **Q2. [The proposed algorithm will obviously increase the decoding time of the coding model. How about a comparison with the COT/RL model like Deepseek R1 for both inference computational cost as well as the code efficiency.]**
>
> While DeepSeek-R1 (with a model size exceeding 600GB) is beyond the hardware constraints of typical academic research environments, we conduct experiments using the distilled variant (the DeepSeek-R1-Distill-Qwen-32B) on the EffiBench dataset. As shown in the results below, although the COT/RL model improves correctness over the base Qwen2.5-Coder-32B model (~59% vs. ~53%, see Table 1 in the main paper), our method achieves further gains in both correctness and efficiency. Also, the computational cost comparison is in line with the trends and analysis reported in Table 2 of the main paper.
>
>
> | Methods                              | ET(AVG) | ET(Median) | NET(Avg) | NET(Median) | NMU  | Time Cost (Median) | Correctness  |
> |--------------------------------------|---------|------------|----------|-------------|------|--------------------|--------------|
> | Default (temperature=0.6, top-p=0.95) | 1.49    | 1.35       | 1.31     | 1.01        | 1.51 | 10.02              | 59.33%       |
> | Perplexity (Best-of-n, n=50)           | 1.51    | 1.32       | 1.22     | 1.02        | 1.53 | 12.12              | 61.22%       |
> | Perplexity (Top P)                     | 1.47    | 1.32       | 1.31     | 0.98        | 1.49 | 9.21               | 58.76%       |
> | EffiLearner                          | 0.63    | 0.68       | 0.83     | 0.73        | 1.32 | 16421.22           | 11.11%       |
> | Self-Debug                           | 0.79    | 0.91       | 0.95     | 0.84        | 1.22 | 22312.09           | 52.55%       |
> | PerfCodeGen                          | 0.68    | 0.78       | 0.89     | 0.84        | 1.01 | 23076.52           | 56.24%       |
> | Ours                                 | 0.11    | 0.64       | 0.71     | 0.63        | 0.93 | 121.84             | 64.12%       |
>
>
> **Q3. [How about the setting of the hyperparameters (\alpha, \beta, \gamma) in General Formulation reward signals?]**
>
> The hyperparameter values used in our main experiments are provided in Appendix C. Below, we present detailed results on hyperparameter tuning using CodeLlama-7B-Instruct on the HumanEval+ dataset. While it is challenging to draw definitive trends given the interplay of three correlated reward components, we generally observe that incorporating all terms leads to the best overall performance. Among them, slightly emphasizing the LLM-based reward ($\beta$) over the perplexity-based term ($\gamma$), e.g., using a weight ratio of 3:2 to 2:1 tends to yield better results. Additionally, modestly increasing the weight of the AST term ($\alpha$) also leads to slight improvements in final performance.  Notably, a broad range of hyperparameter configurations already outperform all baselines, indicating the robustness of our approach. For the final evaluations across different datasets and models, we fix the combination $\alpha = 1.3$, $\beta = 1$, and $\gamma = 0.4$, which achieved the best results in this setting.
> |  $\alpha$ |  $\beta$  |  $\gamma$  | ET (Avg) of | Correctness |
> |:---:|:---:|:---:|:-----------:|:-----------:|
> |  1  |  0  |  0  |     4.89    |    38.27%   |
> |  0  |  1  |  0  |     1.68    |    50.24%   |
> |  0  |  0  |  1  |     2.78    |    45.43%   |
> |  0  |  1  |  1  |     1.67    |    52.55%   |
> | 0.5 |  1  |  1  |     1.64    |    54.49%   |
> |  1  | 0.5 |  1  |     1.72    |    52.25%   |
> |  1  |  1  | 0.5 |     1.61    |    56.63%   |
> |  1  |  1  |  1  |     1.65    |    49.95%   |
> |  1  |  1  | 1.5 |     1.71    |    47.78%   |
> |  1  | 1.5 |  1  |     1.57    |    53.38%   |
> |  1  |  2  |  1  |     1.58    |    55.55%   |
> | 1.5 |  1  |  1  |     1.63    |    51.13%   |
> | 1.4 |  1  | 0.5 |     1.59    |    57.77%   |
> | 1.4 |  1  | 0.4 |     1.62    |    55.58%   |
> | 1.3 |  1  | 0.5 |     1.55    |    59.12%   |
> | 1.3 |  1  | 0.4 |     1.54    |    60.37%   |
> | 1.3 |  1  | 0.3 |     1.57    |    59.08%   |
> | 1.3 |  1  | 0.2 |     1.62    |    58.88%   |
> | 1.3 | 0.7 | 0.4 |     1.62    |    51.21%   |

---

### Decision · Program_Chairs · 2025-09-17

**Decision:**

Accept (poster)

**Comment:**

This paper addresses the important but underexplored problem of generating efficient code with LLMs by introducing an execution-free critique mechanism that guides decoding toward runtime- and memory-efficient solutions. Reviewers agree on its practicality, simplicity, and broad applicability, with consistent performance gains over recent baselines. One common weaknesse raised is the reliance on the accuracy and consistency of the critique LLM. Overall, the consensus is that the paper makes a timely and useful contribution to efficient code generation, and I recommend acceptance.